# Personality traits are consistently associated with blood mitochondrial DNA copy number estimated from genome sequences in two genetic cohort studies

Richard F Oppong[1], Antonio Terracciano[2,3], Martin Picard[4], Yong Qian[1], Thomas J Butler[1], Toshiko Tanaka[1], Ann Zenobia Moore[1], Eleanor M Simonsick[1], Krista Opsahl-Ong[1], Christopher Coletta[5], Angelina R Sutin[6], Myriam Gorospe[5], Susan M Resnick[3], Francesco Cucca[7], Sonja W Scholz[8,9], Bryan J Traynor[9,10], David Schlessinger[5], Luigi Ferrucci[1]*, Jun Ding[1]*

[1]Translational Gerontology Branch, National Institute on Aging, Baltimore, United States; [2]Department of Geriatrics, Florida State University, Tallahassee, United States; [3]Laboratory of Behavioral Neuroscience, National Institute on Aging, Baltimore, United States; [4]Division of Behavioral Medicine, Department of Psychiatry; Merritt Center and Columbia Translational Neuroscience initiative, Department of Neurology, Columbia University Irving Medical Center; New York State Psychiatric Institute, New York, United States; [5]Laboratory of Genetics and Genomics, National Institute on Aging, Baltimore, United States; [6]Department of Behavioral Sciences and Social Medicine, College of Medicine, Florida State University, Tallahassee, United States; [7]Istituto di Ricerca Genetica e Biomedica, Consiglio Nazionale delle Ricerche, Monserrato, Italy; [8]Neurodegenerative Diseases Research Unit, National Institute of Neurological Disorders and Stroke, Bethesda, United States; [9]Department of Neurology, Johns Hopkins University Medical Center, Baltimore, United States; [10]Laboratory of Neurogenetics, National Institute on Aging, Bethesda, United States

*For correspondence:
ferruccilu@grc.nia.nih.gov (LF);
jun.ding@nih.gov (JD)

## Abstract

**Background:** Mitochondrial DNA copy number (mtDNAcn) in tissues and blood can be altered in conditions like diabetes and major depression and may play a role in aging and longevity. However, little is known about the association between mtDNAcn and personality traits linked to emotional states, metabolic health, and longevity. This study tests the hypothesis that blood mtDNAcn is related to personality traits and mediates the association between personality and mortality.

**Methods:** We assessed the big five personality domains and facets using the Revised NEO Personality Inventory (NEO-PI-R), assessed depressive symptoms with the Center for Epidemiologic Studies Depression Scale (CES-D), estimated mtDNAcn levels from whole-genome sequencing, and tracked mortality in participants from the Baltimore Longitudinal Study of Aging. Results were replicated in the SardiNIA Project.

**Results:** We found that mtDNAcn was negatively associated with the Neuroticism domain and its facets and positively associated with facets from the other four domains. The direction and size of the effects were replicated in the SardiNIA cohort and were robust to adjustment for potential confounders in both samples. Consistent with the Neuroticism finding, higher depressive symptoms were associated with lower mtDNAcn. Finally, mtDNAcn mediated the association between personality and mortality risk.

**Conclusions:** To our knowledge, this is the first study to show a replicable association between mtDNAcn and personality. Furthermore, the results support our hypothesis that mtDNAcn is a biomarker of the biological process that explains part of the association between personality and mortality.

**Funding:** Support for this work was provided by the Intramural Research Program of the National Institute on Aging (Z01-AG000693, Z01-AG000970, and Z01-AG000949) and the National Institute of Neurological Disorders and Stroke of the National Institutes of Health. AT was also supported by the National Institute on Aging of the National Institutes of Health Grant R01AG068093.

## Editor's evaluation

This paper makes a comprehensive survey of the relationship between mtDNAcn and the personality dimensions, as well as how and whether they mediate the relationships between personality dimensions and mortality as well as other behavioural measures that may lead to mortality.

## Introduction

Mitochondria are cellular organelles essential for the life of eukaryotic cells (*Duchen, 2004*). They produce adenosine 5'-triphosphate (ATP), the main energetic currency for biological function, by oxidative phosphorylation (OXPHOS) of a number of substrates (*Attardi and Schatz, 1988*). Mitochondria harbor their own circular DNA molecules (mtDNA), which lack histone proteins and bear genes lacking introns (*Attardi and Schatz, 1988*). The proximity of mtDNA to OXPHOS production and its lack of histone protection make it more susceptible to ROS damage, threatening healthy cellular function (*Clay Montier et al., 2009*). The number of copies of mtDNA (mtDNAcn) in tissues and blood leukocytes is considered an imperfect proxy measure of mitochondrial mass that may reflect energetic resilience or hematopoiesis (*Picard, 2021*).

Consequently, reduced mtDNAcn in peripheral blood has been associated with various health outcomes, including renal cell carcinoma (*Xing et al., 2008*), hypertension (*Hägg et al., 2021*), neurodegenerative disease (*Yang et al., 2021*), and diabetes-related conditions, including insulin sensitivity (*Gianotti et al., 2008*; *Song et al., 2001*), fasting plasma glucose level (*Xu et al., 2012*), and diabetic nephropathy (*Al-Kafaji et al., 2018*). High levels of mtDNAcn in tissue and the blood can also occur in conditions of stress and illness such as diabetes (*Moore et al., 2018*) and peripheral artery disease (*McDermott et al., 2018*), suggesting that in some situations of stress, mtDNAcn can be elevated to compensate for low cellular energy (*Picard, 2021*; *Giordano et al., 2014*) and stress adaptation (*Picard and McEwen, 2018a*). For major depression, however, the findings are mixed, with some studies reporting that mtDNAcn has a positive (*Cai et al., 2015*), negative (*Kim et al., 2011*), or null association with depression (*de Sousa et al., 2014*; *He et al., 2014*; *Lindqvist et al., 2018*; *Verhoeven et al., 2018*; *Vyas et al., 2020*).

Since specific personality traits affect individuals' susceptibility to environmental stress (*Ebstrup et al., 2011*) and the development of pathology and premature death, we hypothesized that blood mtDNAcn is a biomarker of stress burden and might mediate the association between personality traits and excess mortality (*Chapman et al., 2020*). To test this hypothesis, we tested the relationship between personality and levels of whole-blood mtDNAcn estimated by whole-genome sequencing and whether mtDNAcn mediated the relation between personality and mortality. To ensure replicability and robustness, we tested these associations in two cohorts, one from the United States and the other from Italy.

To assess personality, we used participants' profiles of the *big five* domains, including the six facets of each domain, measured with the Revised NEO Personality Inventory (NEO-PI-R), a personality measure validated extensively in multiple populations and languages (*Costa and McCrae, 2008*). Personality traits measured with the NEO-PI-R have been associated with stressors such as anxiety and depression (*Hayward et al., 2013*; *Kotov et al., 2010*), cognitive decline (*Caselli et al., 2016*; *Luchetti et al., 2016*), metabolic dysfunction (*Jokela et al., 2014*; *Stephan et al., 2016*), energy levels (*Terracciano et al., 2013*), and physical illness (*Smith and MacKenzie, 2006*). Facets of personality have also been associated with mortality risk (*Chapman et al., 2020*). Compared with behavioral pathways to mortality risk (*Turiano et al., 2015*), there has been less attention on the physiological mechanisms

**eLife digest** Cells are powered by internal structures called mitochondria which have their own DNA molecules. How many copies of mitochondrial DNA blood cells contain is one aspect of mitochondrial health and is considered to provide a good indication of an individual's ability to convert glucose into energy. Consequently, changes in the amount of mitochondrial DNA in the blood are linked to conditions like diabetes and cancer, and have also been associated with aging and mortality.

A set of well-classified personality traits known as 'the Big Five' have also been shown to affect energy levels and the longevity of individuals. However, it remained unclear if there is a relationship between these characteristics and the number of copies of mitochondrial DNA in the blood.

To investigate, Oppong et al. used a specialized test to assess the personality traits of participants from two separate cohorts: Baltimore Longitudinal Study of Ageing and the SardiNIA Project. The genomic sequence of each person was then analyzed to calculate the amount of mitochondrial DNA in their blood, and their mortality was recorded based on whether they were alive or dead multiple years later.

Oppong et al. found that low levels of mitochondrial DNA were linked with high scores in neuroticism (a trait typically associated with anxiety, depression, and self-doubt). Further statistical tests revealed that mitochondrial DNA levels mediate the relationship between a person's personality and their risk of death.

These findings suggest that personality traits impact the number of mitochondrial DNA molecules in a person's blood, which, in turn, influences how long they are likely to live. However, further work is needed to find out what causes this effect.

that explain the well-known association between personality and mortality risk. As mentioned above, lower mtDNAcn has been associated with both stressful situations and higher mortality (*Mengel-From et al., 2014*). We thus address the critical novel question by testing whether mtDNAcn mediates the association between personality traits and mortality risk.

The goal of this study was to investigate the association between blood mtDNAcn and personality traits and facets assessed by the NEO-PI-R in two epidemiological studies: the Baltimore Longitudinal Study of Aging (BLSA) (*Kuo et al., 2020*; *Stone and Norris, 1966*) and the SardiNIA study (*Pilia et al., 2006*). Using mediation analysis, we further tested the hypothesis that mtDNAcn is a biomarker in the biological process that connects personality characteristics with mortality.

## Methods
### Study cohorts' description
#### Baltimore Longitudinal Study of Aging
The BLSA is an ongoing scientific study that aims to understand risk factors and pathways that cause a decline in physical and cognitive function with aging. The study began in 1958 and is run by the National Institute on Aging Intramural Research Program and enrolls healthy volunteers. Blood samples were obtained from 955 individuals, and DNA was extracted from a red-cell-free buffy coat and sequenced on a HiSeq X Ten sequencer using 150 base-pair, paired-end cycles (version 2.5 chemistry, Illumina), as described elsewhere (*Chia et al., 2021*). For genetic homogeneity, we retained only participants with white ancestry for further analysis downstream. We further excluded participants who had developed Alzheimer's disease (AD) from further analysis because their NEO-PI-R profiles are likely changed by clinical dementia (*Islam et al., 2019*). Therefore, 722 participants (mean age 75, ranging from 48 to 100, 48% were women; *Table 1*) remained for downstream analysis. All participants gave written informed consent, with protocols approved by the Institutional Review Board of the Intramural Research Program of the National Institutes of Health (protocol number 03-AG-0325).

#### SardiNIA Longitudinal Study
SardiNIA (*Pilia et al., 2006*) is a longitudinal study of human aging and genetics of quantitative traits that has followed a genetically isolated population on the Italian island of Sardinia since 2000. The data used for this study consist of 2077 participants who had their DNA from whole blood sequenced

**Table 1.** Demography of study cohorts.

| | BLSA (n = 722) | SardiNIA (n = 587) |
|---|---|---|
| | n/%/mean (SD) | n/%/mean (SD) |
| Female | 48.1% | 62.2% |
| Age (years) | 74.88 (10.86) | 57.28 (13.24) |
| Education (years) | 17.68 (2.50) | 7.33 (3.96) |
| Big five personality traits | | |
| Neuroticism | 45.35 (8.25) | 55.46 (8.36) |
| Extraversion | 50.60 (10.10) | 46.80 (8.04) |
| Openness | 52.59 (10.55) | 44.35 (9.08) |
| Agreeableness | 51.88 (9.51) | 47.55 (8.55) |
| Conscientiousness | 51.91 (9.31) | 49.20 (8.79) |
| Measures of depressive symptoms | | |
| CES-D continuous | 5.47 (5.66) | 11.66 (7.79) |
| CES-D binary cutoff 16 | 6.6% | 23.8% |
| CES-D binary cutoff 20 | 2.6% | 13.9% |

BLSA, Baltimore Longitudinal Study of Aging; CES-D, Center for Epidemiologic Studies Depression Scale.

using Illumina Genome Analyzer IIx and Illumina HiSeq 2000 instruments. We further excluded one of each pair of participants with a genetic relationship above the level of second-degree cousins. A total of 587 participants (mean age 57, ranging from 15 to 96, 62% women; *Table 1*) remained for further downstream analysis. All participants gave written informed consent, with protocols approved by the Institutional Review Board of the Intramural Research Program of the National Institutes of Health (protocol number 04-AG-N317).

## Phenotype measures and definition

Personality was assessed using the NEO-PI-R (*Costa and McCrae, 2008*). The NEO-PI-R is an inventory of 240 items that examines an individual's thoughts, feelings, and behaviors (*Costa and McCrae, 2008*). Each item is answered on a five-point Likert scale ranging from 'strongly disagree' to 'strongly agree.' Responses to these items are scored to assess the *big five* personality traits (the five domains of personality traits): Neuroticism (N), Extraversion (E), Openness (O), Agreeableness (A), and Conscientiousness (C). In addition to the five domains, the NEO-PI-R also assesses six facets within each domain, for a total of 30 facets (e.g., for the Neuroticism domain, the following six facets are assessed: Anxiety [N1], Angry Hostility [N2], Depression [N3], Self-Consciousness [N4], Impulsiveness [N5], Vulnerability [N6]). Scores are standardized using a normative population to generate standardized *T scores* (mean = 50, SD = 10) (*Costa and McCrae, 2008*). These *T scores* were used in the analysis.

BLSA participants completed the NEO-PI-R at each clinic visit (four assessment points on average), and we used the average across all assessment points for the downstream analysis. We did this because barring onset and progression of clinical dementia, which causes a drastic change in NEO-PI-R profiles (*Terracciano and Sutin, 2019*), NEO-PI-R scores are quite stable in adults over long periods (*Costa and McCrae, 1988*). Averaging the NEO-PI-R scores over multiple visits improves the BLSA analysis by reducing the standard deviation of the effects estimates and therefore increases statistical power. SardiNIA participants completed the NEO-PI-R once, and we used this one-time measure for downstream analysis. An extensive literature supports the validity and reliability of the NEO-PI-R scales in the United States and other countries, including Italy (*McCrae and Terracciano, 2005*; *Terracciano, 2003*). In both the BLSA and SardiNIA, the five traits have high internal consistency (alpha > 0.80), and in the BLSA, the test–retest stability of the five traits ranges from 0.78 to 0.83 over a follow-up of 10 years (*Costa et al., 2007*; *Terracciano et al., 2006*; *Appendix 1—figure 5*). The correlations between the NEO-PI-R traits are shown in *Appendix 1—figure 6*.

We also analyzed in both studies the measure of depressive symptoms using the Center for Epidemiologic Studies Depression Scale (CES-D) (*Radloff, 1977*). The CES-D is a 20-item instrument assessing depressive mood and behavior over the past week. Responses to these statements are graded on a 4-point Likert scale ranging from 'rarely or none of the time' to 'most or all of the time.' The responses are scored from 0 to 3 and summed across the 20 items to assess depressive symptoms (range 0–60). In general populations, a CES-D score of 16 is the clinical cutoff used to classify depressive symptoms (*Beekman et al., 1997*; *Delitala et al., 2016*). However, among older populations, a cutoff score of 20 is used to improve the accuracy of the diagnosis of major depression (*Milaneschi et al., 2011*; *Penninx et al., 1998*). We, therefore, analyzed two binary clinical cutoffs of CES-D (16 and 20 for significant depressive symptoms) and continuous scores in both studies. CES-D in both studies was assessed at a single time point.

## mtDNA copy number estimation

The whole-blood mtDNAcn of each study participant was estimated from whole-genome sequence information using *fastMitoCalc* (*Qian et al., 2017*). This software implements the formula below, as described by *Ding et al., 2015*.

$$mtDNA\ copy\ number\ per\ cell = \frac{mtDNA\ average\ coverage}{autosomal\ DNA\ average\ coverage} \times 2$$

The above formula was derived from the fact that average sequencing coverage should be proportional to DNA copy number for autosomal DNA and mtDNA. Since there are two copies of autosomal DNA in each cell, the mtDNAcn is inferred as a ratio of average mtDNA coverage to autosomal DNA coverage multiplied by two. The average coverage of mtDNA and autosomal DNA (*Appendix 1—figure 7*) is obtained using sequence alignment with *SAMtools* (*Li et al., 2009*).

## Statistical analysis

We performed initial simple linear regression of mtDNAcn values on variables such as age, sex, sequence coverage (the average autosomal coverage for each study participant), platelet count, and white blood cell (WBC) parameters, including WBC count, percentages of the major leukocytes: lymphocytes, neutrophils, eosinophils, monocytes, and basophils. These initial assessments helped us determine which variables should be included as covariates in subsequent multiple linear regression models of mtDNAcn on NEO-PI-R traits.

As expected from previous findings reviewed in *Picard, 2021*, age, sex, WBC count, platelet count, and percentages of neutrophils, lymphocytes, and basophils were significantly associated with mtDNAcn (p-value<0.0001). Sequence coverage and percentage of eosinophils were nominally associated with mtDNAcn (p-value<0.05), while percentage of monocytes was marginally associated with mtDNAcn (p-value=0.0821). Therefore, all regression analyses relating mtDNAcn with personality traits or depression were adjusted for age, sex, WBC and platelet count, percent neutrophils, lymphocytes, basophils, and eosinophils, and sequence coverage.

Standardized values were used in regression analyses to make effect sizes more comparable across the two study cohorts. Therefore, all reported NEO-PI-R associations are expressed as the number of standard deviations change in mtDNAcn in response to higher NEO-PI-R trait values. A random-effect meta-analysis was used to obtain the combined estimates of effects and 95% confidence interval (CI) of each NEO-PI-R/ CES-D trait on mtDNAcn using the standardized effect sizes obtained from the two study cohorts.

## Personality, mtDNAcn, and mortality risk

In previous studies, the *vulnerability* facet of Neuroticism, the *activity* facet of *Extraversion*, and the *self-discipline* and *competence* facets of Conscientiousness were associated with mortality risk, with *vulnerability* (increased risk) and *self-discipline* (reduced risk) showing the strongest associations (*Chapman et al., 2020*). To confirm these findings in the literature, we classified BLSA and SardiNIA participants according to *vulnerability* and *self-discipline* scores below and above the median level, and cross-classified participants into four groups by combinations of these two dichotomous variables (see *Appendix 1—figure 8* for more details). We tested their association with mtDNAcn using the low-vulnerability/high self-discipline group as a reference. We also used the Cox proportional hazards model to test for their association with mortality, adjusting for age and sex.

In subsequent analyses, we computed a personality-mortality index (PMI) using data from the four NEO-PI-R facets (*vulnerability*, *activity*, *self-discipline*, and *competence*) that have been previously shown as significantly associated with mortality by *Chapman et al., 2020*. For each of the four traits, we assigned a score of 0 or 1 to each individual based on the median value and the direction of the association of the trait with mortality. For example, individuals with *vulnerability* measures lower than the median received a score of 1 for the trait, and individuals with *self-discipline* measures higher than the median received a score of 1. The final PMI is the sum of the four scores for the four traits. The index ranged between 0 and 4, where a score of 0 means an individual has a high *vulnerability*, low *self-discipline,* low *activity,* and low *competence* (i.e., a poor PMI and potentially highest mortality risk). A score of 4 represents low *vulnerability*, high *self-discipline*, high *activity,* and high *competence*, indicating a favorable PMI and potentially lowest mortality risk. Scores of 1, 2, and 3 are various levels of improved PMI. We compared the mtDNAcn among the five indices (0, 1, 2, 3, and 4) of the PMI. We also tested for an association between PMI as a continuous variable and mtDNAcn, and death (a binary variable that classified study participants as dead or alive).

Finally, we used R package lavaan (version 0.6-7) to perform a mediation analysis to test whether mtDNAcn mediates the association between personality (i.e., PMI) and mortality risk (a binary variable describing the dead or alive status of study participants). In general, there are three steps in a mediation analysis to show a potential mediator variable mediates the relationship between an independent variable and a dependent variable. Briefly, the three steps in our specific analysis are (1) a total effect model, which regresses mortality (dependent variable) on PMI (independent variable) to confirm that PMI is a significant predictor of mortality; (2) a mediator model, which regresses mtDNAcn (mediator) on PMI to confirm that PMI is a significant predictor of mtDNAcn; and (3) an outcome model, which regresses mortality on both mtDNAcn and PMI to confirm that (a) mtDNAcn is a significant predictor of mortality and (b) the strength of the previously significant effect of PMI in step (1) is now greatly reduced (i.e., the effect of PMI on mortality is largely indirect via mtDNAcn). In our analysis, the significance of the indirect effect was estimated using 5000 bootstrap samples.

## Results

### Relationship of sex, age, and WBC parameters with mtDNAcn

In both cohorts, mtDNAcn was negatively associated with age and male biological sex (*Appendix 1—table 1*). Furthermore, higher mtDNAcn was associated with lower WBC count and lower neutrophil percentage and was also associated with higher platelet count, lymphocyte percentage, eosinophil percentage, monocyte percentage, and basophil percentage (*Appendix 1—table 1*), consistent with previous findings (*Picard, 2021*). After adjusting for the effect of sex, sequence coverage (mean sequence coverage of 35.8× and 3.8×, respectively, for BLSA and SardiNIA), platelet count, and WBC parameters, there remained a significant, albeit modest, inverse association of mtDNAcn with age in both study cohorts (*Figure 1* and *Appendix 1—table 1*). For the BLSA cohort, on average, every 1-year increase in age was associated with a 0.014 standard deviation decrease in mtDNAcn (about 3.89% decrease in mtDNAcn per decade of life, p-value = $2.5 \times 10^{-5}$), while in the SardiNIA cohort, there was a 0.0065 standard deviation decline in mtDNAcn for every year increase in age (about 1.36% decrease in mtDNAcn per decade of life, p-value=0.028). After adjusting for all the other covariates, women still had a significantly higher mtDNAcn than men (*Figure 1* and *Appendix 1—table 1*).

### Association of personality traits with mtDNA copy number

To test the association between NEO-PI-R traits and mtDNAcn in each study cohort, we regressed mtDNAcn values on each personality trait, adjusting for age, sex, sequence coverage, platelet count, and WBC parameters. In both samples, mtDNAcn was associated with lower Neuroticism and higher Extraversion, Openness, Agreeableness, and Conscientiousness (*Figure 2*). We then performed a random-effects meta-analysis and used a false discovery rate (FDR)-corrected p-value threshold of 0.01. Among the 35 personality traits (5 domains, 30 facets) tested, 9 (5 facets in the Neuroticism domain and 4 from Extraversion, Agreeableness, and Conscientiousness) were associated significantly with mtDNAcn even after the stringent FDR correction. *Table 2* shows the individual study and meta-analysis results for the Neuroticism domain, while *Appendix 1—table 2* provides detailed results for all 35 NEO-PI-R traits.

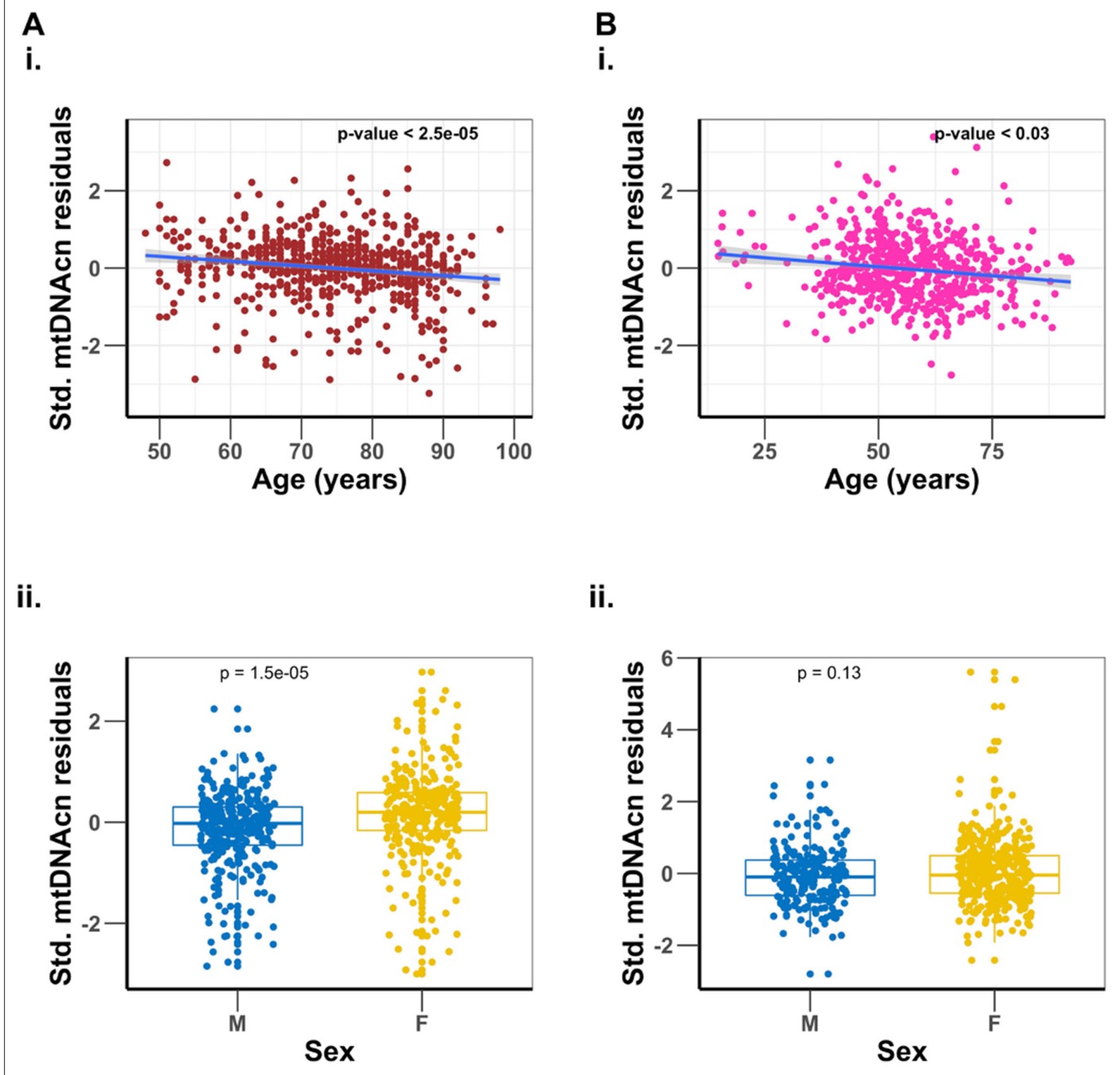

**Figure 1.** Association of mitochondrial DNA copy number (mtDNAcn) with age, and sex differences. Panel (**A**) is the Baltimore Longitudinal Study of Aging (BLSA) cohort, and panel (**B**) is the SardiNIA cohort. Plots (**i**) are regression of mtDNAcn residuals on age after adjusting for sex, sequence coverage, platelets count, and white blood cell parameters. Plots (**ii**) are comparisons of mtDNAcn residuals after adjusting for age, sequence coverage, platelets count, and white blood cell parameters between males (M) and females (F), with *t*-test p-values shown.

The Neuroticism domain showed the most significant associations with mtDNAcn. The overall Neuroticism domain and four out of six facets were associated significantly (FDR p-value≤0.01) with mtDNAcn (*Table 2*). The four facets were anxiety, angry hostility, depression, and vulnerability. Of note, facet self-consciousness was associated with mtDNAcn with an FDR-corrected p-value of 0.013. Notably, all neuroticism traits were associated negatively with mtDNAcn, with higher trait values corresponding to lower mtDNAcn values. Furthermore, the higher-order Neuroticism domain was the personality trait associated with mtDNAcn with the most significant p-value (FDR p-value=0.0015).

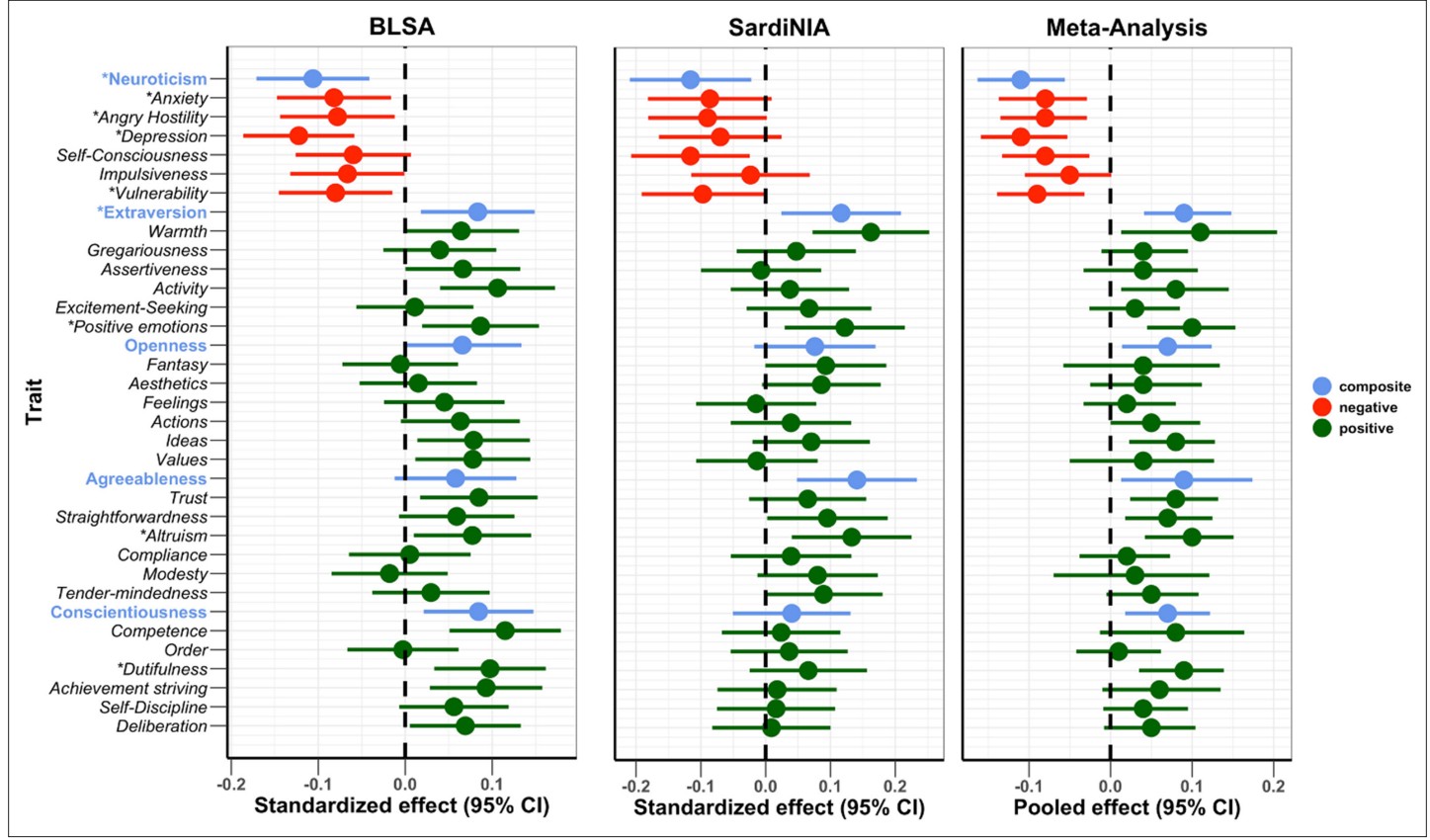

**Figure 2.** Association of Revised NEO Personality Inventory (NEO-PI-R) traits with mitochondrial DNA copy number (mtDNAcn). Standardized/pooled effects with 95% CI of NEO-PI-R traits after adjusting for the effects of age, sex, sequence coverage, platelets count, and white blood cell parameters. Color coding is green for positive personality types, red for negative personality types, and blue for the composite scores for the big five NEO-PI-R domains. * Traits with false discovery rate (FDR)-corrected meta-analysis p-values≤0.01.

The Neuroticism domain also showed highly consistent associations between the two study cohorts, with $I^2$ (a measure of inconsistency among individual studies) equal to 0 for Neuroticism and its facets. *Appendix 1—figure 1* shows the effect sizes for the Neuroticism domain and facets.

The overall Extraversion domain, the *positive emotions* facet in the Extraversion domain, the *altruism* facet in the Agreeableness domain, and the *dutifulness* facet in the Conscientiousness domain were associated with mtDNAcn with FDR-corrected p-values<0.01 (*Appendix 1—table 2*). These are all positive associations, with higher trait values corresponding to higher mtDNAcn values. No facets in the Openness domain were significantly associated with mtDNAcn.

*Appendix 1—figure 2* compares the standardized effect sizes of the NEO-PI-R traits between the two study cohorts. The results show remarkable concordance between the two study cohorts, with the data points clustering close to the diagonal line. In a sensitivity analysis evaluating the robustness of these findings to covariate adjustments, we compared the standardized effect sizes of the personality traits using models with and without adjustment for platelet count and WBC parameters (*Appendix 1—figure 3*). As most of the points were closely aligned to the diagonal line, we concluded that adjusting for platelet count and WBC parameters did not significantly impact the results.

## Association of CES-D with mtDNAcn

In both cohorts, we also regressed mtDNAcn on both continuous and binary CES-D values, adjusting for the effect of other covariates. Higher CES-D scores, measured as a continuous variable, were associated with lower mtDNAcn. Participants with depressive symptoms that exceeded the CES-D cutoff of either 16 or 20 had 6.06 or 11.57% lower mtDNAcn than those with no depressive symptoms, respectively (*Table 2*). Remarkably, in both cohorts, the effect sizes almost doubled when moving from moderate depressive symptoms (CES-D binary cutoff 16) to significant depressive symptoms (CES-D

**Table 2.** Association of Neuroticism domain with mitochondrial DNA copy number (mtDNAcn).

| Domain and facets | BLSA β (p-value) | SardiNIA β (p-value) | Pooled β (p-value)* | $I^2$ % [p of Q] |
|---|---|---|---|---|
| Neuroticism | –0.106 (0.001) | –0.116 (0.01) | –0.11 (0.0015) | 0% [0.87] |
| *Anxiety* | –0.082 (0.01) | –0.086 (0.08) | –0.08 (0.010) | 0% [0.94] |
| *Angry hostility* | –0.078 (0.02) | –0.090 (0.054) | –0.08 (0.010) | 0% [0.84] |
| *Depression* | –0.122 (0.0002) | –0.070 (0.1) | –0.11 (0.0015) | 0% [0.37] |
| *Self-consciousness* | –0.060 (0.08) | –0.116 (0.01) | –0.08 (0.013) | 0% [0.33] |
| *Impulsiveness* | –0.067 (0.046) | –0.023 (0.6) | –0.05 (0.099) | 0% [0.45] |
| *Vulnerability* | –0.080 (0.02) | –0.097 (0.04) | –0.09 (0.0087) | 0% [0.77] |
| | | | | |
| Measures of depressive symptoms | | | | |
| *CES-D continuous* | –0.122 (0.0002) | –0.077 (0.13) | –0.11 (0.00008) | 0% [0.46] |
| *CES-D binary cutoff 16* | –0.260 (0.051) | –0.180 (0.13) | –0.22 (0.015) | 0% [0.65] |
| *CES-D binary cutoff 20* | –0.591 (0.005) | –0.343 (0.02) | –0.42 (0.0004) | 0% [0.34] |
| | | | | |
| PMI | | | | |
| *Personality-mortality index* | 0.092 (0.005) | 0.124 (0.007) | 0.10 (0.0001) | 0% [0.57] |

BLSA, Baltimore Longitudinal Study of Aging; CES-D, Center for Epidemiologic Studies Depression Scale; PMI, personality-mortality index; FDR, false discovery rate.

*Pooled effect p-values of Neuroticism domain and facets are FDR-corrected for 35 tests for 35 personality traits. Information for other personality domains and facets are provided in *Appendix 1—table 2*.

binary cutoff 20). The random-effects meta-analysis supported both the magnitude and direction of the effects with highly significant association p-values. There was also high consistency across the two cohorts for the continuous and binary CES-D traits, with $I^2$ equal to 0.

## Personality, mtDNAcn, and mortality risk

We further studied the relationship between mtDNAcn and personality facets known to be associated with mortality risk. *Appendix 1—figure 4* compares the mtDNAcn among four distinct personality types (see 'Methods' for details) based on the *vulnerability* facet of Neuroticism and *self-discipline* facet of Conscientiousness, which were the two individual facets most strongly associated with mortality (*Chapman et al., 2020*). Comparing the mtDNAcn between the four personality types in the BLSA cohort, we observed that the personality type with the perceived highest mortality risk (high vulnerability low self-discipline [HVLD]) had the lowest mtDNAcn among the four personality types. The HVLD personality type had a mean mtDNAcn value 0.23 standard deviations lower than the type with a perceived lowest mortality risk ( low vulnerability high self-discipline [LVHD]) (p-value<0.0065), indirectly supporting an association between blood mtDNAcn and mortality risk. A more detailed description and interpretation of the results can be found in Appendix 1.

## mtDNAcn mediates the association between personality and mortality risk

The above preliminary observation that mtDNAcn was associated with mortality-related personality traits led us to a more detailed investigation of the relationship between personality, blood mtDNAcn, and mortality. In particular, we further extended our analysis to include four facets (*vulnerability*, *self-discipline*, *activity*, and *competence*) by creating a PMI (see 'Methods' for details) and performed a mediation analysis.

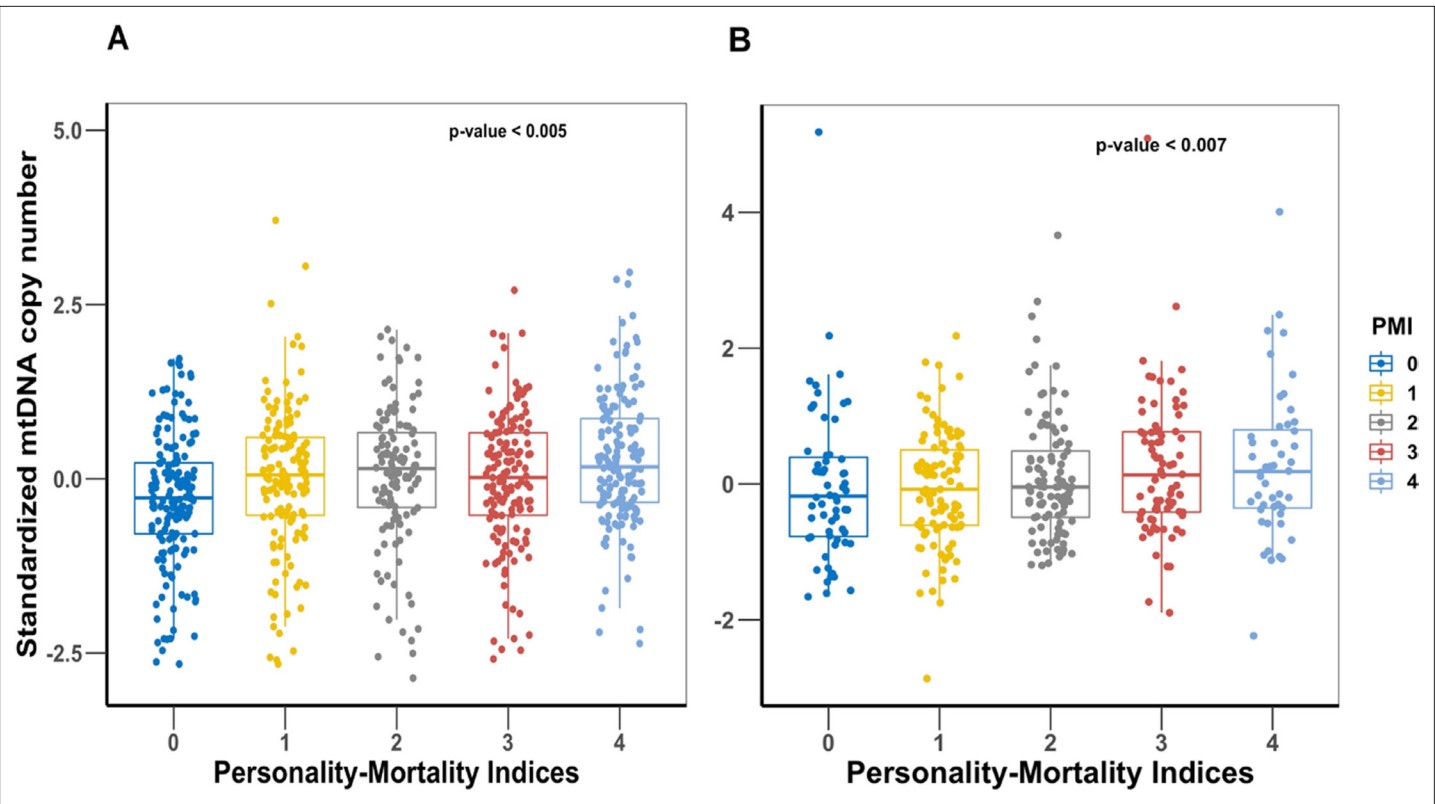

**Figure 3.** Comparison of mitochondrial DNA copy number (mtDNAcn) values among the five personality-mortality indices created from the *vulnerability* facet of Neuroticism, *activity* facet of Extraversion, and *self-discipline* and *competence* facets of Conscientiousness. Panel (**A**) is Baltimore Longitudinal Study of Aging (BLSA) cohort, and panel (**B**) is SardiNIA cohort. The personality-mortality index with the lowest mortality risk is 4, and the index with the highest mortality risk is index 0. p-Values of linear regression of mtDNAcn and the personality-mortality index are shown on the plots.

First, we performed a Cox proportional hazards analysis for the four NEO-PI-R facets (***Appendix 1—table 3***). The analysis showed that after adjusting for age and sex, all four facets are significantly associated with increased risk of mortality (vulnerability) or reduced risk of mortality (activity, self-discipline, and competence). For the PMI analysis, we observed an increasing mtDNAcn when moving from index 0 (worst PMI) to index 4 (excellent PMI) in both study cohorts (***Figure 3***). Analyzing the PMI as a continuous variable showed that higher trait values were associated with higher mtDNAcn in both cohorts (***Table 2***). We confirmed the association between PMI and mortality by finding that the participants had significantly longer survival time with increasing PMI from 0 to 4 (***Figure 4***).

Given the significant associations of mtDNAcn and specific personality traits with mortality, we then tested the hypothesis that the association between PMI and mortality may be mediated through mtDNAcn. We used structural equation modeling to test the indirect effect of PMI on mortality via mtDNAcn. The results are shown in ***Table 3*** and ***Figure 4***. We observed that for BLSA, PMI was a significant predictor of mortality. The indirect effect estimate showed that the effect of PMI on mortality was fully mediated via mtDNAcn. We tested the significance of the indirect effect using 5000 bootstrap samples, which yielded a significant p-value=0.002 (estimate = –0.097). Similar results were obtained for SardiNIA, although they were not statistically significant, likely due to limited power because the Sardinian cohort is younger, and the mortality data are limited compared to the BLSA cohort. Also, the Sardinian analysis had reduced statistical power compared to the BLSA analysis because the personality measures are averaged across multiple visits for BLSA, which decreases the standard deviation (increasing the reliability) of effect estimates.

## Discussion

An extensive literature has reported the association of blood mtDNAcn with various psychological outcomes such as major depression (***Cai et al., 2015***; ***Kim et al., 2011***), autism (***Giulivi et al., 2010***),

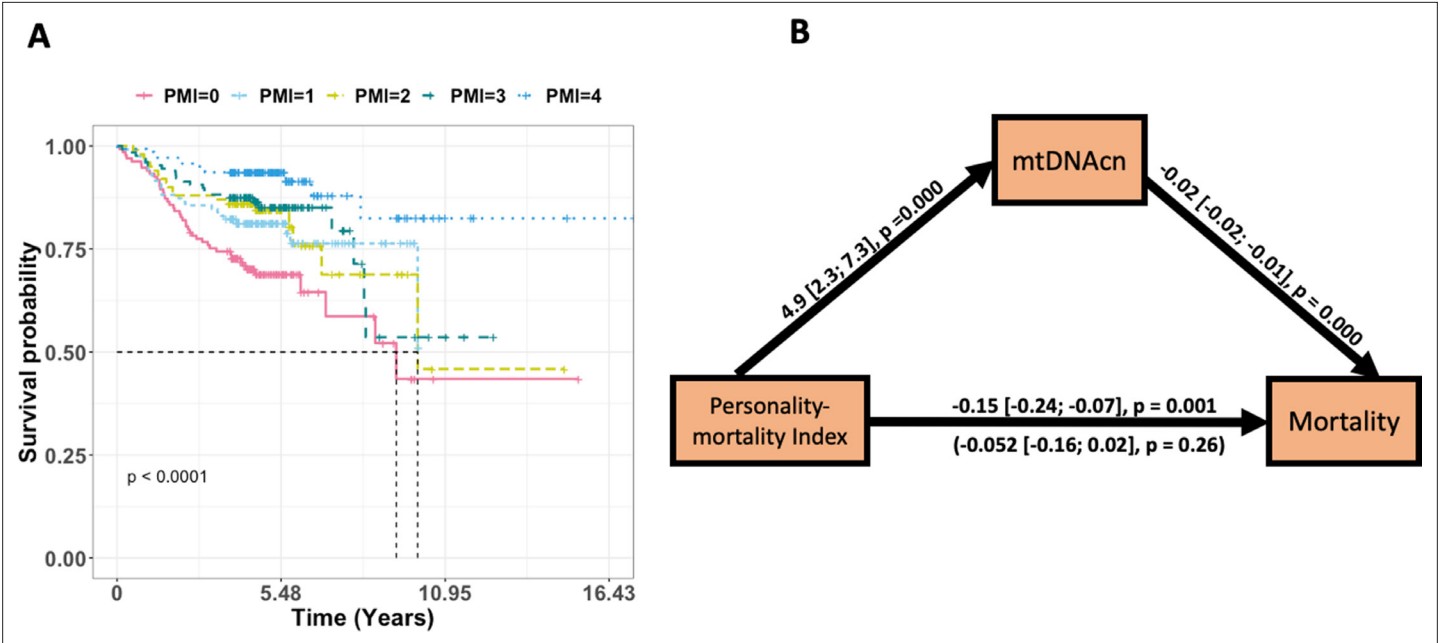

**Figure 4.** The effect of personality-mortality index (PMI) on mortality is significantly mediated via mitochondrial DNA copy number (mtDNAcn). Panel (**A**) is a Kaplan–Meier plot of time to death of Baltimore Longitudinal Study of Aging (BLSA) participants in the five PMIs. Panel (**B**) shows the regression coefficients between PMI (predictor), mortality (outcome), and mtDNAcn (mediator) for the BLSA study. The effects of age and sex are adjusted for in the models. The indirect effect is (4.89) × (–0.02) = –0.097. We tested for the significance of the indirect effect using 5000 bootstrap samples and it was statistically significant (p-value=0.002). See **Table 3** for more details.

bipolar disorder (**Chang et al., 2014**), and stress (**Bersani et al., 2016**; **Boeck et al., 2016**; **Tyrka et al., 2016**). Our study adds new information to this literature by reporting an association between mtDNAcn and personality traits. Our results show that higher mtDNAcn is significantly associated with a lower level of Neuroticism and higher levels of Extraversion, Openness, Agreeableness, and Conscientiousness in study participants. We demonstrate the robustness of these associations by showing that the direction and size of the effects are robust to adjustment for potential confounders such as age, sex, platelet count, and WBC parameters and are replicated across two cohorts. Consistent with Neuroticism, participants with more depressive symptoms had lower mtDNAcn than those without depressive symptoms. Finally, we report an effect of personality on mortality and demonstrate that this effect is partially mediated via mtDNAcn.

mtDNAcn is a biomarker of energetic health and cellular aging (**Picard and McEwen, 2018a**). The mechanism that controls the levels of mtDNAcn in tissues, blood, and plasma is not fully understood,

**Table 3.** Testing the direct and indirect effects of personality-mortality index and mitochondrial DNA copy number (mtDNAcn) on mortality.

| | BLSA (n = 721) | SardiNIA (n = 395) |
|---|---|---|
| Total effect | Estimate (p-value) | Estimate (p-value) |
| Personality-mortality index | –0.149 (0.001) | –0.022 (0.808) |
| Full effects | | |
| Personality-mortality index | –0.052 (0.260) | –0.019 (0.895) |
| mtDNAcn | –0.020 (0.000) | –0.001 (0.966) |
| Indirect effect | Estimate [nonparametric bootstrap 95% CI] | Estimate [nonparametric bootstrap 95% CI] |
| mtDNAcn | –0.097 [–0.143; –0.03] | –0.004 [–0.169; 0.189] |

BLSA, Baltimore Longitudinal Study of Aging.

and it is unclear whether the change of blood mtDNAcn reflects mitochondrial biogenesis, immune health, and/or hematopoiesis in the bone marrow (*Picard, 2021*). Increased mtDNAcn have been shown to have a different and potentially opposite indication depending on the underlying health condition (*Moore et al., 2018*). In population-based cohorts such as those involved in this study, increased mtDNAcn is generally positively correlated with mitochondria mass and thus is generally associated with higher oxidative capacity. Having a large pool of well-functioning mitochondria can positively influence health and generate signals that contribute to stress adaptation (*Picard and McEwen, 2018a*). Consistent with this interpretation, we found that the mtDNAcn levels were higher in younger participants, and our primary findings indicated that individuals with higher mtDNAcn levels had lower Neuroticism and were at lower risk of depression and lower risk of mortality. In adverse health conditions, however, it should be noted that increased mtDNAcn may not correlate with function (*Moore et al., 2018*; *Wei et al., 2001*). Rather, it may indicate a compensatory upregulation of mtDNAcn (*Picard, 2021*), a phenomenon where mtDNAcn is increased to compensate for low-energy cellular environment due to poor mitochondrial energetics (*Picard, 2021*; *Giordano et al., 2014*).

Personality traits influence behaviors, emotional states, and how individuals deal with stress. For example, personality traits are associated with how individuals assess stressful life circumstances (*Ebstrup et al., 2011*) and what coping responses to the stressors (i.e., neuroendocrine responses, immune functioning and inflammation, cardiovascular responses) are elicited (*Schneiderman et al., 2005*). Given these associations with stress and health-related behaviors, it is not surprising that personality traits are also related to better health, function, and longevity (*Hayward et al., 2013*; *Caselli et al., 2016*; *Smith and MacKenzie, 2006*; *Terracciano and Sutin, 2019*; *Bogg and Roberts, 2004*; *Sutin et al., 2019*; *Canada et al., 2021*). We observed a positive association between mtDNAcn and personality traits that are also related to health-relevant habits and conditions, such as nonsmoking (*Malouff et al., 2006*), healthy diet and obesity (*Jokela et al., 2013*; *Mõttus et al., 2013*), exercise and physical function (*Kekäläinen et al., 2020*), and preventive medical screenings (*Aschwanden et al., 2019*). Thus, indirect pathways may exist to link personality and health-related behaviors to changes in mitochondrial biology. Our finding of an association between mtDNAcn and personality expands this broader literature on personality traits and health.

Reduced mtDNAcn may reflect reduced mitochondrial mass or loss of mtDNA due to mitochondrial dysfunction (*Picard, 2021*). Dysfunctional mitochondria that are not appropriately removed by mitophagy produce excess ROS, which causes oxidative stress. As a result of damage caused by oxidative stress, mitochondria release mitochondrial antigenic molecules called damage-associated molecular patterns (DAMPs) into the cytoplasm (*Picard and McEwen, 2018a*; *Zampino et al., 2020*). These released DAMPs contain mtDNA, which are unmethylated and thus act as antigenic fragments and elicit an innate immune response through toll-like receptors (*Zhang et al., 2010*). This triggers an inflammatory response through cytokine release (*Picard and McEwen, 2018a*; *Zampino et al., 2020*). Prolonged inflammation, when unresolved, can lead to damage to cells and tissues and has been associated with a number of chronic diseases, including cardiovascular diseases (*Lopez-Candales et al., 2017*), depression (*Maydych, 2019*), and neurodegeneration (*Missiroli et al., 2020*). Our study shows that low mtDNAcn is associated with higher levels of Neuroticism and facets of Neuroticism, including depression. Of note, data from the SardiNIA sample indicates that Neuroticism is associated with higher interleukin-6 (IL-6) and C-reactive protein (CRP) (*Sutin et al., 2010*). We observed a similar relationship between mtDNAcn and CES-D (depressive symptoms), and the effect becomes even more pronounced (almost doubling in effect size) when considering significant depressive symptoms (setting CES-D binary cutoff at 20 instead of at 16). *Kim et al., 2011* report a similar negative association between mtDNAcn and depression. However, other studies like *Cai et al., 2015* and *Tyrka et al., 2016* report a positive association of mtDNAcn with major depression, and several studies have found null associations (*de Sousa et al., 2014*; *He et al., 2014*; *Lindqvist et al., 2018*; *Verhoeven et al., 2018*; *Vyas et al., 2020*). One explanation for these inconsistencies could be the heterogeneity of the depression phenotyping: the depression facet of the Neuroticism domain is a trait measure, and the CES-D measures symptoms, but neither is a diagnostic instrument for major depression. It should also be noted that although some past studies assayed for mtDNAcn from whole blood, they did not adjust for the effect of platelets, WBCs, and other leukocyte subgroups on mtDNAcn as we do in this study. mtDNAcn estimates differ among the various blood cell types (*Rausser et al., 2021*), and

this significantly impacts whole-blood mtDNAcn estimates and should be adjusted for in association models (also see discussion below).

mtDNAcn and personality are both predictors of mortality (*Chapman et al., 2020*; *Mengel-From et al., 2014*; *Ashar et al., 2015*), thus making our finding of an association between the two even more intriguing. Understanding which variable affects the other in driving the effect on mortality is a significant step forward, but alternative models are similarly plausible. To test the hypothesis that mtDNAcn mediates the association between personality and mortality, we performed a causal mediation analysis. In this article, we provide the first evidence of mtDNAcn significantly mediating the association between personality (including three facets capturing positive psychosocial states [activity, self-discipline, and competence] and an emotional stressor [vulnerability]) and mortality. Advances in mitochondrial research have presented the mitochondrion as a target of stress (*Picard and McEwen, 2018b*), and more recently, as a potential biological mediator of the stress-disease cascade (*Picard and McEwen, 2018a*). *Picard and McEwen, 2018a* provide a conceptual model for mitochondrial stress pathophysiology, outlining how mitochondria interact with psychosocial factors and stressors to influence health and lifespan outcomes. The underlying mechanism for this mediation is still an area of active research and is not fully understood, but the significance of our results remains clear: mtDNAcn is one mechanism in the pathway that connects personality and mortality.

This study also shows that mtDNAcn was negatively associated with age and male biological sex, which is consistent with previous reports (*Ding et al., 2015*). Also, higher mtDNAcn was significantly associated with higher levels of platelet count and percentages of lymphocytes, eosinophils, monocytes, and basophils. However, higher mtDNAcn was associated with lower WBC count and lower neutrophil percentage. Neutrophils have relatively lower mtDNAcn compared to other leukocytes (*Rausser et al., 2021*), and they form the largest proportion of WBC, which may explain the negative association of WBC count with mtDNAcn. A similar negative association between mtDNAcn and WBC count was observed by *Liu et al., 2020*. This study and other recent publications *Hurtado-Roca et al., 2016*; *Knez et al., 2016*; *Shim et al., 2020* have highlighted the importance of adjusting for WBC parameters and platelet count when testing the association between traits and mtDNAcn measured in whole blood.

The primary strength of this study is the use of two independent study cohorts, one in the United States and the other in Europe: the top associations not only had significant p-values in both studies but also had very similar estimated effect sizes. Furthermore, in both cohorts, the associations of the personality traits and mtDNAcn ranked high when put in a broader context of mtDNAcn associations with other traits. The consistency of the results with the previous literature also clearly demonstrates the reproducibility of this study. The way mtDNAcn is estimated using whole-genome sequence information from the two study cohorts is also seen as the gold standard going forward in this line of research (*Filograna et al., 2021*), with major mitochondria consortia such as TopMed (*Liu et al., 2020*) employing the same approach. A limitation of this study was the one-time assessment of mtDNAcn, which precluded a longitudinal analysis of changes in mtDNAcn. Personality was assessed only once in the SardiNIA cohort, but personality is relatively stable in adults over long periods (*Costa and McCrae, 1988*; *Terracciano et al., 2006*); therefore, not much information may be lost in a cross-sectional analysis of personality. Additionally, we note that the formula for mtDNA copy number calculates a weighted average (weights determined by the proportion of leukocyte populations) of mtDNA copy number per cell across all leukocytes; each cell has two copies of nuclear DNA that provide the reference for the calculation. However, this does not take into account the platelet content of the buffy coat. Because each platelet has about 1.6 copies of mtDNA (*Hurtado-Roca et al., 2016*; *Urata et al., 2008*) but does not contain nuclear DNA, the formula somewhat overestimates mtDNA copy number (*Rausser et al., 2021*). The platelet content of buffy coat may also vary, representing a limitation of this and other studies of mtDNA in blood DNA material (*Picard, 2021*). Because of this limitation, papers published on mtDNA copy number typically use both platelets count and WBC counts as covariates in any models testing the association between mtDNA copy number and quantitative traits. We also used this conventional approach in our analyses. Another drawback of this study is the difference in the distribution of age and mtDNAcn across the two cohorts. The differences in the average mtDNAcn between the two cohorts can be explained by the gap in the time at which sequencing was done; almost a decade passed between the time the two cohorts were sequenced. Despite these

limitations, the associations between mtDNAcn and personality traits are clear and consistent across the two study cohorts.

In this study, we showed a significant association between mtDNAcn and neuroticism traits that were replicated in two independent cohorts. Other larger cohorts (e.g., UK Biobank) have also collected certain personality information (e.g., questions about neuroticism), as well as whole-genome sequencing data (which makes estimating mtDNAcn feasible). Therefore, trying to replicate our findings in such larger cohorts will be a natural extension of this study.

In conclusion, we provide the first evidence of an association between mtDNAcn and personality. These results may provide further evidence of the link between mtDNAcn and psychosocial stress and suggest that blood mtDNAcn may correlate with changes in mitochondrial biology under stress. These initial findings will require further exploration to reveal the potential causal pathways between personality and mtDNAcn. We also provide novel insights into the effect of personality on mortality by showing that this effect is mediated through mtDNAcn.

## Acknowledgements

Support for this work was provided by the Intramural Research Program of the National Institute on Aging (Z01-AG000693, Z01-AG000970, and Z01-AG000949) and the National Institute of Neurological Disorders and Stroke of the National Institutes of Health. AT was also supported by the National Institute on Aging of the National Institutes of Health Grant R01AG068093.

## Additional information

### Competing interests

Sonja W Scholz: participates as a Scientific advisory board member for Lewy body dementia association. The author has no other competing interests to declare. Bryan J Traynor: has indicated several grants, patents, positions and interests in his individual ICMJE form, but he has done it in the interest of transparency and none of his disclosures is related to the content of the current manuscript. David Schlessinger: is a consultant for Elixirgen Therapeutics, Inc The author has no other competing interests to declare. The other authors declare that no competing interests exist.

### Funding

| Funder | Grant reference number | Author |
|---|---|---|
| National Institute on Aging | Intramural Research Program | Bryan J Traynor Bryan J Traynor |
| National Institute of Neurological Disorders and Stroke | Intramural Research Program | Sonja W Scholz |
| National Institute on Aging | Grant R01AG068093 | Antonio Terracciano |

The funders had no role in study design, data collection and interpretation, or the decision to submit the work for publication.

### Author contributions

Richard F Oppong, Conceptualization, Formal analysis, Validation, Investigation, Methodology, Writing – original draft; Antonio Terracciano, Martin Picard, Investigation, Methodology, Writing – original draft; Yong Qian, Thomas J Butler, Software, Writing – review and editing; Toshiko Tanaka, Ann Zenobia Moore, Eleanor M Simonsick, Christopher Coletta, Angelina R Sutin, Data curation, Writing – review and editing; Krista Opsahl-Ong, Formal analysis, Writing – review and editing; Myriam Gorospe, Data curation, Funding acquisition, Investigation, Writing – review and editing; Susan M Resnick, Francesco Cucca, Sonja W Scholz, Bryan J Traynor, Data curation, Investigation, Writing – review and editing; David Schlessinger, Conceptualization, Data curation, Funding acquisition, Investigation, Writing – original draft; Luigi Ferrucci, Conceptualization, Data curation, Formal analysis, Supervision, Funding acquisition, Investigation, Methodology, Writing – original draft; Jun

Ding, Conceptualization, Formal analysis, Supervision, Validation, Methodology, Writing – original draft

## Author ORCIDs
Richard F Oppong http://orcid.org/0000-0002-3244-6694
Antonio Terracciano http://orcid.org/0000-0001-5799-8885
Martin Picard http://orcid.org/0000-0003-2835-0478
Toshiko Tanaka http://orcid.org/0000-0002-4161-3829
Luigi Ferrucci http://orcid.org/0000-0002-6273-1613
Jun Ding http://orcid.org/0000-0002-4302-5027

## Ethics
All participants of the two cohort studies gave written informed consent, with protocols approved by the Institutional Review Board of the Intramural Research Program of the National Institutes of Health (protocol numbers 03-AG-0325 and 04-AG-N317 for BLSA and SardiNIA, respectively).

## Decision letter and Author response
Decision letter https://doi.org/10.7554/eLife.77806.sa1
Author response https://doi.org/10.7554/eLife.77806.sa2

## Additional files

### Supplementary files
• Transparent reporting form

• Reporting standard 1. A checklist of items that should be included in reports of observational studies.

• Source code 1. R code for the analysis on mtDNAcn and personality.

### Data availability
Researchers interested in using BLSA data (e.g., sequencing or phenotypic data) need to submit an analysis plan to a review committee organized by the National Institute on Aging (A detailed guideline for the submission of an analysis plan can be found at https://www.blsa.nih.gov/how-apply). Researchers will get the access once the proposed analysis plans are reviewed and approved. Aggregated genetic and phenotypic data from the SardiNIA study are deposited in the European Genome-phenome Archive (EGA) (https://ega-archive.org/studies/phs000338). In accord with European Union regulations, disaggregated data, including sequence/genotyping data of individuals, cannot be made public, but investigators can request specific analyses by sending a one-page proposal description to Dr. F. Cucca (fcucca@uniss.it), the P.I. of the study. After screening by an internal committee, the requested analyses are customarily done by staff of IRGB and the results provided to the requestor. The R code for the analyses performed in this paper is provided as a Supplementary file (source code file: R code for the analysis on mtDNAcn and personality).

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

# Appendix 1

## Supplementary information text

Personality,          mtDNAcn,          and          mortality          risk

We studied the relationship between mtDNAcn and personality facets known to be associated with mortality risk. *Appendix 1—figure 8* compares the mtDNAcn among four distinct personality types (see 'Methods' for details) created from the vulnerability facet of Neuroticism and self-discipline facet of Conscientiousness, which were the two individual facets most strongly associated with mortality. When comparing the mtDNAcn between the four personality types in the BLSA cohort, we observed that the personality type with the perceived highest mortality risk (HVLD) had the lowest mtDNAcn among the four personality types. The type with a perceived lowest mortality risk (LVHD) had a mean mtDNAcn value that is 0.23 standard deviations higher than the highest mortality type (HVLD) (p-value<0.0065). As a comparison of effect size relative to the slope of mtDNAcn and age after adjusting for all covariates, this difference in mtDNAcn between personality types corresponds to the difference in mtDNAcn observed across 0.6 years (about 7 months) of life. A similar mtDNAcn difference was observed in the SardiNIA cohort: the mean mtDNAcn difference between the two personality types was 0.21 standard deviations. This difference corresponds to the difference in mtDNAcn observed across 0.3 years (a little over 3 months) of life.

## PMI does not mediate the association between mtDNAcn and mortality risk

We used causal mediation analysis to test the indirect effect of mtDNAcn on mortality via PMI as a mediator. The indirect effect estimate showed that the effect of mtDNAcn on mortality is not mediated through PMI. We tested for the significance of the indirect effect using 5000 bootstrap samples, and it was not statistically significant (p-values=0.355 and 0.963, respectively, for BLSA and SardiNIA). See *Appendix 1—table 4* for more details.

## Menopause status does not affect the association between mtDNAcn and personality

Given the mean age of 57 years in the SardiNIA cohort, we investigated menopausal status as a potential confounder in the association between mtDNAcn and personality. We found that there was no significant difference between the mtDNAcn of premenopausal and menopausal women (p-value=0.56) (*Appendix 1—figure 9*). The results show that menopause had no effect on mtDNAcn in the SardiNIA cohort. Also, we ran two association models between mtDNAcn and NEO--PI-R traits in women, with and without adjustment for the effect of menopause status (both models adjusted for the effect of age, sequence coverage, WBC, platelet count, and percentages of major types of leukocytes). The results are shown in *Appendix 1—table 5*. The results show that menopause status does not affect the association between mtDNAcn and personality: the effect sizes and p-values were almost the same in both models for all the traits.

## mtDNAcn is not influenced by mitochondrial haplogroups

We investigated the effect of the major mitochondrial haplogroups present in our dataset. There were 17 haplogroups in total, and we excluded 7 haplogroups that were extremely rare (present in less than five study participants). The most common haplogroup was H (frequency of about 42%). *Appendix 1—figure 10* shows the relative mtDNAcn in each haplogroup. mtDNAcn has a similar distribution among the different haplogroups, although slight variations in mtDNAcn were observed among them. We further investigated whether these mtDNAcn variations among the haplogroups were statistically significant by testing for the association of each haplogroup versus all other haplogroups with mtDNAcn. The results are shown in *Appendix 1—table 6*. Among the 10 haplogroups present in more than five study participants, none was significantly associated with mtDNAcn. This finding suggests that none of the haplogroups present in our study participants will confound our results.

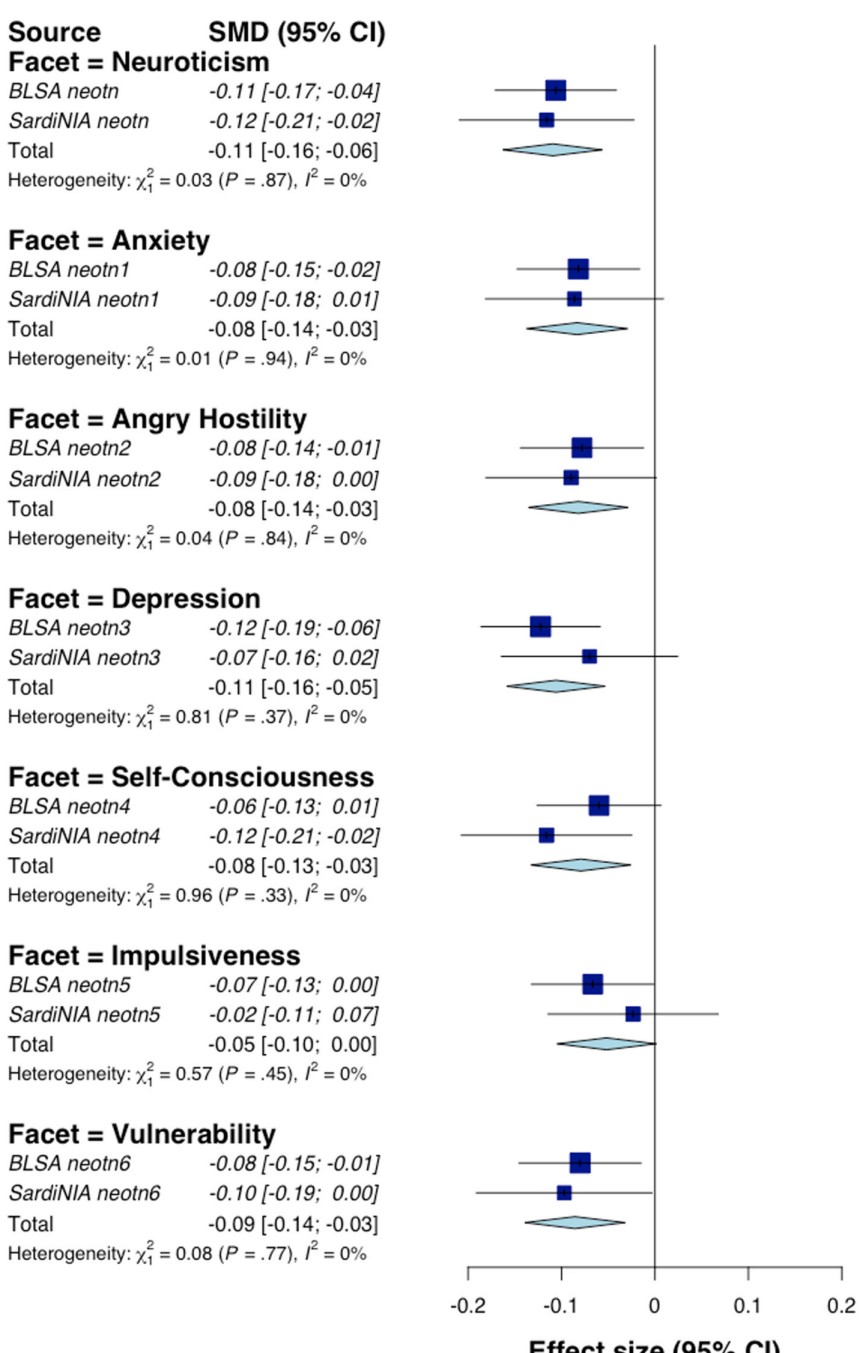

**Appendix 1—figure 1.** Random-effect meta-analysis of Neuroticism domain of Revised NEO Personality Inventory (NEO-PI-R).

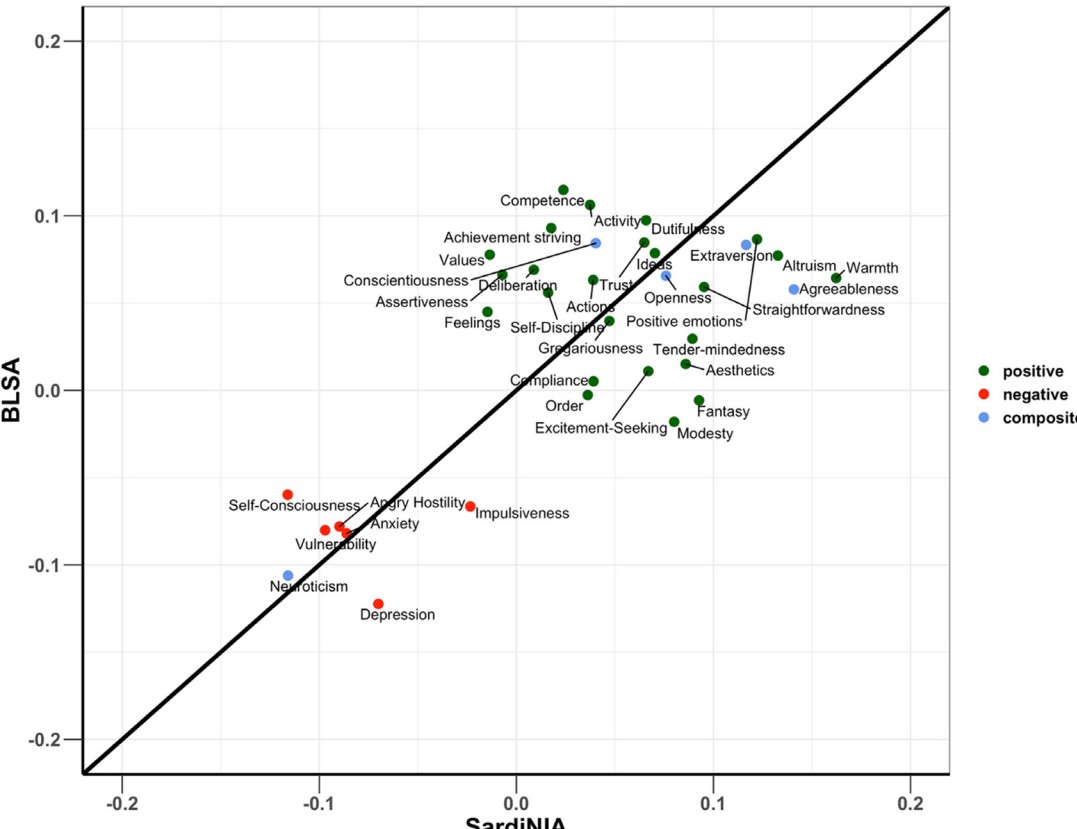

**Appendix 1—figure 2.** Comparison of effect sizes of Revised NEO Personality Inventory (NEO-PI-R) traits between the two cohorts. Regression model adjusted for the effects of age, sex, sequence coverage, and white blood cell parameters. Color coding is green for positive personality types, red for negative personality types, and blue for the composite scores for the big five NEO-PI-R domains.

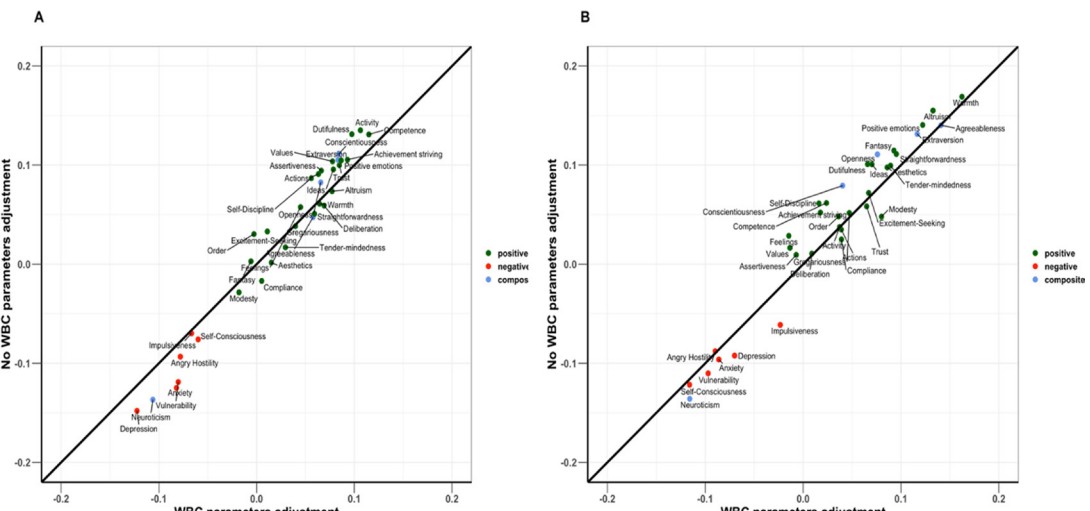

**Appendix 1—figure 3.** Comparison of effect sizes of Revised NEO Personality Inventory (NEO-PI-R) traits between models with white blood cell (WBC) parameters adjustments and with no adjustments. Panel (**A**) is Baltimore Longitudinal Study of Aging (BLSA) cohort, and panel (**B**) is SardiNIA cohort. Color coding is green for positive personality types, red for negative personality types, and blue for the composite scores for the big five NEO-PI-R domains.

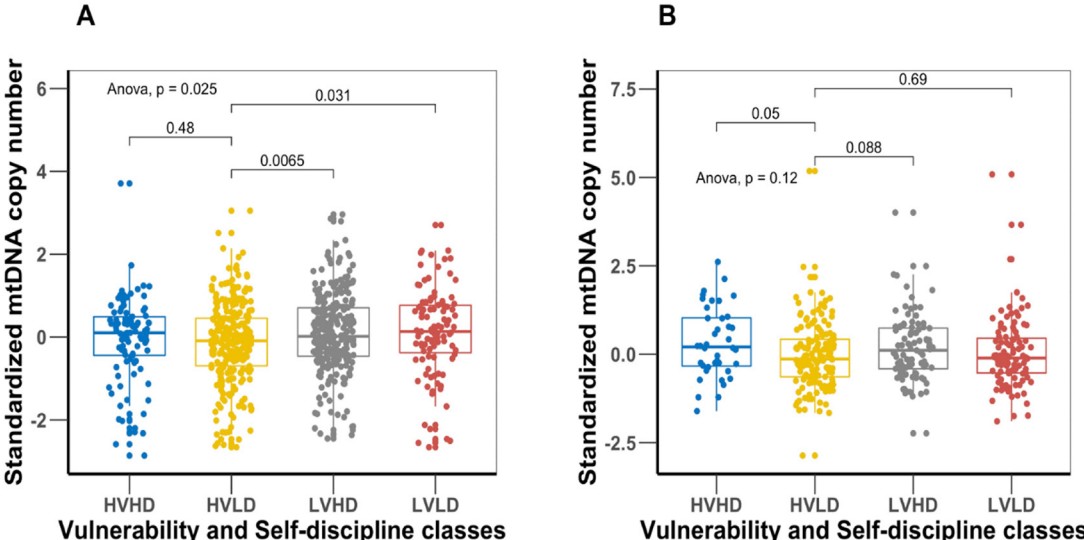

**Appendix 1—figure 4.** Comparison of mitochondrial DNA copy number (mtDNAcn) values among the four distinct personality types created from the vulnerability facet of Neuroticism and self-discipline facet of Conscientiousness. Panel (**A**) is Baltimore Longitudinal Study of Aging (BLSA) cohort, and panel (**B**) is SardiNIA cohort. The personality type with lowest mortality risk (low vulnerability high self-discipline [LVHD]) has a significantly higher mean mtDNAcn value than the highest mortality risk type (high vulnerability low self-discipline [HVLD]).

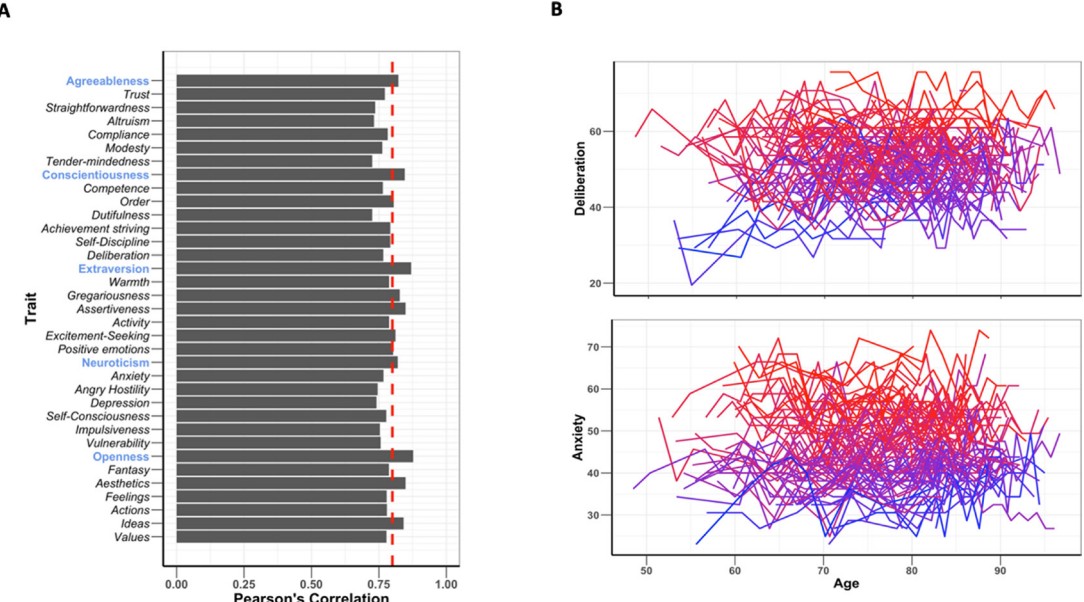

**Appendix 1—figure 5.** Stability of Revised NEO Personality Inventory (NEO-PI-R) traits across multiple visits. (**A**) Test–retest stability of the NEO-PI-R traits. Red dash line is Pearson's correlation of 0.8. All the traits have a high test–retest reliability. (**B**) The stability of two randomly selected personality traits (Deliberation and Activity) across multiple measurement visits. Plot color represents starting trait values; blue shows individual's lower initial trait values, and red shows higher initial trait values. The plot shows that subsequent trait measurement tends to stabilize around the initial measurement values.

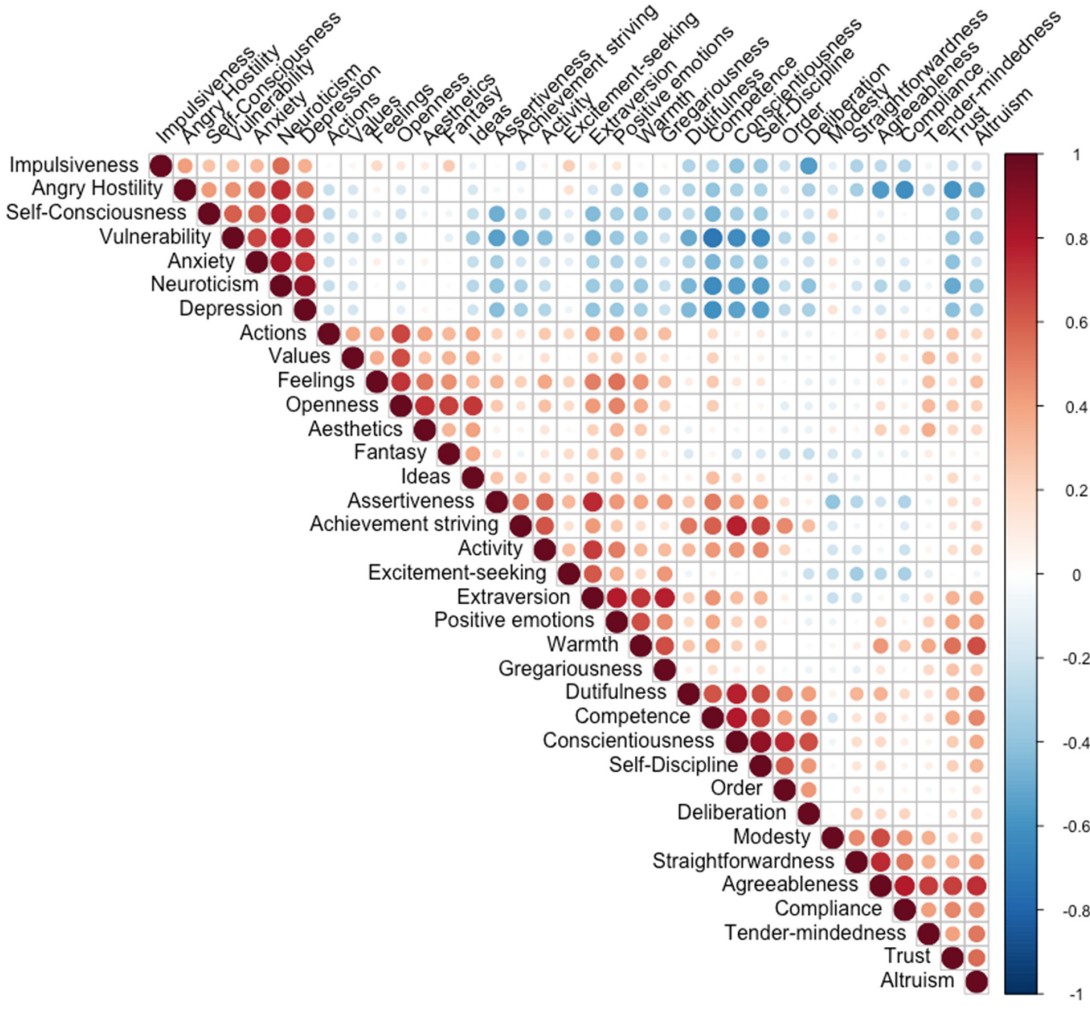

**Appendix 1—figure 6.** Correlation between Revised NEO Personality Inventory (NEO-PI-R) traits. The color intensity and size of circles are indications of the size and significance of the correlation.

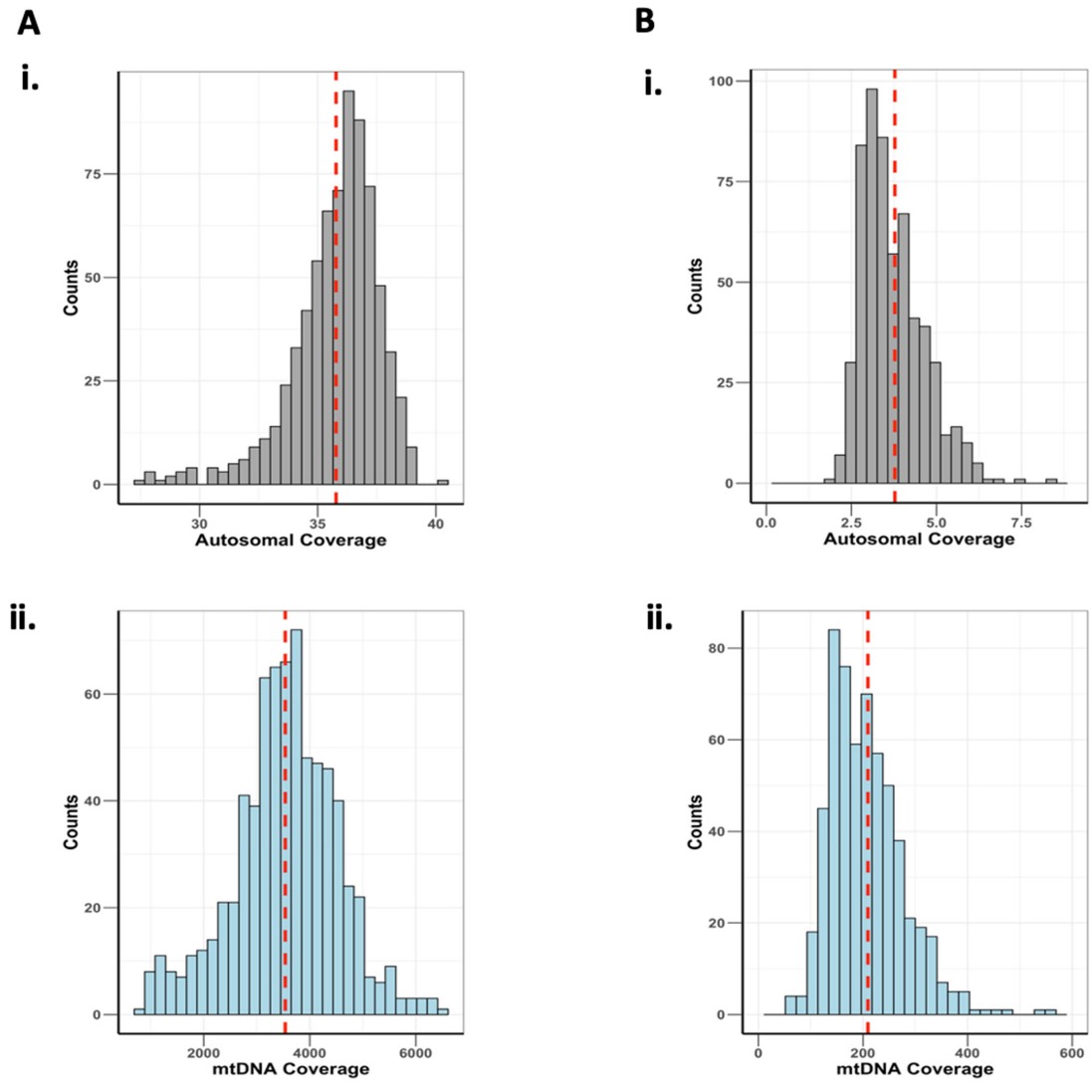

**Appendix 1—figure 7.** Histograms of autosomal and mitochondrial DNA (mtDNA) sequencing coverages for (**A**) Baltimore Longitudinal Study of Aging (BLSA) and (**B**) SardiNIA.

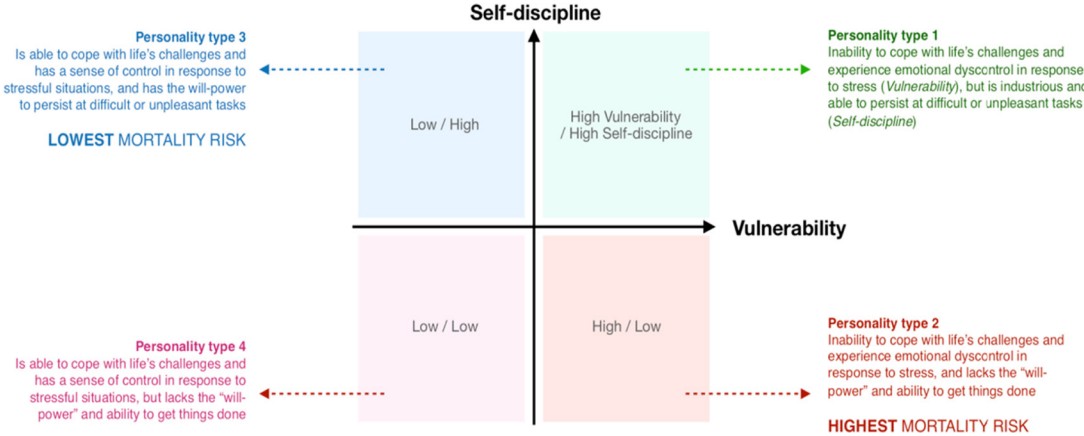

**Appendix 1—figure 8.** Four distinct personality types are created from the vulnerability facet of Neuroticism and the self-discipline facet of Conscientiousness.

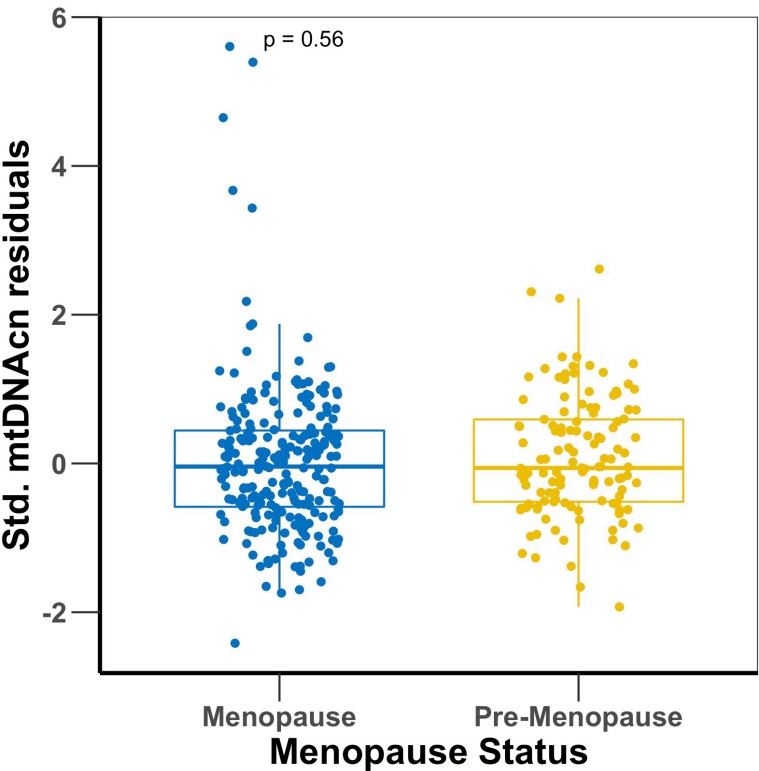

**Appendix 1—figure 9.** Mitochondrial DNA copy number (mtDNAcn) comparison between menopausal and premenopausal women in SardiNIA.

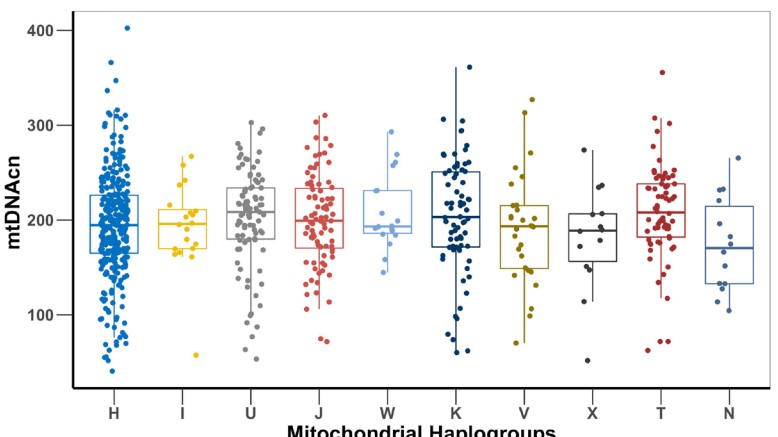

**Appendix 1—figure 10.** Mitochondrial DNA copy number (mtDNAcn) comparison between the major mitochondrial haplogroups present in more than five study participants.

**Appendix 1—table 1.** Association of age, sex, and white blood cell (WBC) parameters with mitochondrial DNA copy number (mtDNAcn).

| | β (p-value) of model adjusted | | | |
|---|---|---|---|---|
| | No adjustment | | Age, sex, coverage, and WBC parameters | |
| Trait | BLSA | SardiNIA | BLSA | SardiNIA |
| Age | –0.025 (2.85e-13) | –0.0093 (0.003) | –0.014 (2.52e-5) | –0.0065 (0.028) |

*Appendix 1—table 1 Continued on next page*

*Appendix 1—table 1 Continued*

|  | β (p-value) of model adjusted | | | |
|---|---|---|---|---|
| Sex | –0.61 (8.14e-17) | –0.25 (0.004) | –0.36 (1.21e-6) | –0.13 (0.11) |
| Platelet count | 0.004 (1.31e-7) | 0.0007 (0.30) | 0.003 (3.71e-6) | 0.001 (0.063) |
| WBC count | –0.13 (2.35e-10) | –0.14 (1.37e-8) | –0.097 (1.48e-6) | –0.08 (0.002) |
| Lymphocytes percent | 0.04 (5.5e-25) | 0.042 (5.19e-16) | –0.033 (0.23) | 0.073 (0.55) |
| Neutrophils percent | –0.041 (1.47e-28) | –0.041 (6.19e-18) | –0.063 (0.021) | 0.036 (0.77) |
| Eosinophils percent | 0.037 (0.031) | 0.043 (0.06) | –0.045 (0.16) | 0.076 (0.55) |
| Monocytes percent | 0.022 (0.082) | 0.062 (0.002) | –0.014 (0.57) | 0.082 (0.52) |
| Basophils percent | 0.44 (8.54e-5) | 0.11 (0.38) | 0.044 (0.69) | –0.076 (0.55) |

BLSA, Baltimore Longitudinal Study of Aging.

**Appendix 1—table 2.** Association of Revised NEO Personality Inventory (NEO-PI-R) domains with mitochondrial DNA copy number (mtDNAcn).

| NEO-PI-R domains | BLSA β (p-value) | SardiNIA β (p-value) | Meta-analysis | | | |
|---|---|---|---|---|---|---|
|  |  |  | Pooled β | Raw p-value | FDR adj. p* | $I^2$ % [p of Q] |
| Neuroticism | –0.106 (0.001) | –0.116 (0.01) | –0.11 | 0.00006 | **0.0015** | 0% [0.87] |
| *Anxiety* | –0.082 (0.01) | –0.086 (0.08) | –0.08 | 0.002 | **0.010** | 0% [0.94] |
| *Angry hostility* | –0.078 (0.02) | –0.090 (0.054) | –0.08 | 0.003 | **0.010** | 0% [0.84] |
| *Depression* | –0.122 (0.0002) | –0.070 (0.1) | –0.11 | 0.00008 | **0.0015** | 0% [0.37] |
| *Self-consciousness* | –0.060 (0.08) | –0.116 (0.01) | –0.08 | 0.004 | 0.013 | 0% [0.33] |
| *Impulsiveness* | –0.067 (0.046) | –0.023 (0.6) | –0.05 | 0.06 | 0.099 | 0% [0.45] |
| *Vulnerability* | –0.080 (0.02) | –0.097 (0.04) | –0.09 | 0.002 | **0.0087** | 0% [0.77] |
| Extraversion | 0.083 (0.01) | 0.117 (0.01) | 0.09 | 0.0005 | **0.0036** | 0% [0.56] |
| *Warmth* | 0.064 (0.06) | 0.162 (0.0004) | 0.11 | 0.03 | 0.051 | 66% [0.09] |
| *Gregariousness* | 0.040 (0.2) | 0.047 (0.3) | 0.04 | 0.1 | 0.16 | 0% [0.90] |
| *Assertiveness* | 0.066 (0.048) | –0.007 (0.9) | 0.04 | 0.3 | 0.36 | 38% [0.21] |
| *Activity* | 0.106 (0.002) | 0.037 (0.4) | 0.08 | 0.02 | 0.040 | 30% [0.23] |
| *Excitement seeking* | 0.011 (0.7) | 0.067 (0.2) | 0.03 | 0.3 | 0.36 | 0% [0.35] |
| *Positive emotions* | 0.087 (0.01) | 0.122 (0.01) | 0.10 | 0.0004 | **0.0036** | 0% [0.54] |
| Openness | 0.066 (0.06) | 0.076 (0.1) | 0.07 | 0.01 | 0.032 | 0% [0.86] |
| *Fantasy* | –0.006 (0.9) | 0.093 (0.051) | 0.04 | 0.4 | 0.48 | 65% [0.09] |
| *Aesthetics* | 0.015 (0.7) | 0.086 (0.06) | 0.04 | 0.2 | 0.27 | 34% [0.22] |
| *Feelings* | 0.045 (0.2) | –0.015 (0.8) | 0.02 | 0.4 | 0.47 | 2% [0.31] |
| *Actions* | 0.063 (0.07) | 0.039 (0.4) | 0.05 | 0.051 | 0.094 | 0% [0.68] |
| *Ideas* | 0.079 (0.02) | 0.070 (0.1) | 0.08 | 0.005 | 0.014 | 0% [0.88] |
| *Values* | 0.078 (0.02) | –0.014 (0.8) | 0.04 | 0.4 | 0.46 | 59% [0.12] |
| Agreeableness | 0.058 (0.1) | 0.141 (0.003) | 0.09 | 0.02 | 0.047 | 49% [0.16] |
| *Trust* | 0.085 (0.014) | 0.065 (0.2) | 0.08 | 0.005 | 0.014 | 0% [0.73] |
| *Straightforwardness* | 0.059 (0.08) | 0.095 (0.04) | 0.07 | 0.009 | 0.024 | 0% [0.54] |
| *Altruism* | 0.077 (0.02) | 0.133 (0.005) | 0.10 | 0.0005 | **0.0036** | 0% [0.34] |

*Appendix 1—table 2 Continued on next page*

*Appendix 1—table 2 Continued*

| NEO-PI-R domains | BLSA β (p-value) | SardiNIA β (p-value) | Meta-analysis | | | |
| | | | Pooled β | Raw p-value | FDR adj. p* | I² % [p of Q] |
|---|---|---|---|---|---|---|
| *Compliance* | 0.005 (0.9) | 0.039 (0.4) | 0.02 | 0.5 | 0.57 | 0% [0.57] |
| *Modesty* | –0.018 (0.6) | 0.080 (0.09) | 0.03 | 0.6 | 0.62 | 65% [0.09] |
| *Tendermindedness* | 0.030 (0.4) | 0.089 (0.053) | 0.05 | 0.07 | 0.12 | 7% [0.30] |
| Conscientiousness | 0.084 (0.009) | 0.040 (0.4) | 0.07 | 0.008 | 0.022 | 0% [0.43] |
| *Competence* | 0.115 (0.0004) | 0.024 (0.6) | 0.08 | 0.09 | 0.14 | 61% [0.11] |
| *Order* | –0.003 (0.9) | 0.036 (0.4) | 0.01 | 0.7 | 0.69 | 0% [0.49] |
| *Dutifulness* | 0.097 (0.003) | 0.066 (0.2) | 0.09 | 0.001 | **0.0064** | 0% [0.57] |
| *Achievement striving* | 0.093 (0.005) | 0.018 (0.7) | 0.06 | 0.09 | 0.14 | 42% [0.19] |
| *Self-discipline* | 0.056 (0.08) | 0.016 (0.7) | 0.04 | 0.1 | 0.14 | 0% [0.48] |
| *Deliberation* | 0.069 (0.03) | 0.009 (0.8) | 0.05 | 0.09 | 0.14 | 11% [0.29] |

| Personality and mortality | BLSA Md (p-value) | SardiNIA Md (p-value) | Pooled Md | Raw p-value | FDR adj. p | I2 % [p of Q] |
|---|---|---|---|---|---|---|
| HVLD vs. LVHD | 0.234 (0.006) | 0.209 (0.09) | 0.226 | 0.0013 | - | 0% [0.87] |

BLSA, Baltimore Longitudinal Study of Aging; HVLD, high vulnerability low self-discipline; LVHD, low vulnerability high self-discipline; FDR, false discovery rate.
*Entries with FDR-corrected p-values≤0.01 are labeled as bold.

**Appendix 1—table 3.** Cox proportional hazards analysis of four Revised NEO Personality Inventory (NEO-PI-R) traits used to compute personality-mortality index (PMI).
Models adjusted for the effect of age and sex.

| | Coefficient | Hazard ratio | p-Value |
|---|---|---|---|
| Vulnerability | 0.037 | 1.038 | 0.0006 |
| Activity | –0.036 | 0.946 | 0.0004 |
| Self-discipline | –0.028 | 0.972 | 0.002 |
| Competence | –0.041 | 0.959 | 2.28e-5 |

**Appendix 1—table 4.** Testing the direct and indirect effects of mitochondrial DNA copy number (mtDNAcn) and personality-mortality index (PMI) on mortality.

| | BLSA (n = 721) | SardiNIA (n = 395) |
|---|---|---|
| **Total effect** | Estimate (p-value) | Estimate (p-value) |
| mtDNAcn | –0.010 (0.001) | 0.003 (0.574) |
| **Full effects** | | |
| mtDNAcn | –0.010 (0.000) | 0.003 (0.673) |
| PMI | –0.118 (0.340) | –0.028 (0.959) |
| **Indirect effect** | Estimate (nonparametric bootstrap p-value) | Estimate (nonparametric bootstrap p-value) |
| PMI | 0.000 (0.355) | 0.000 (0.963) |

BLSA, Baltimore Longitudinal Study of Aging.

**Appendix 1—table 5.** Association of Revised NEO Personality Inventory (NEO-PI-R) domains with mitochondrial DNA copy number (mtDNAcn) in SardiNIA women, with and without adjustment for menopause status.

Both models adjusted for the effect of age, sequence coverage, white blood cell (WBC), platelet count, and percentages of major leukocytes.

| NEO-PI-R domains | β (p-value) of model | |
| --- | --- | --- |
| | Not adjusted for menopause status | Adjusted for menopause status |
| Neuroticism | –0.013 (0.074) | –0.013 (0.077) |
| Anxiety | –0.007 (0.316) | –0.007 (0.323) |
| Angry hostility | –0.010 (0.123) | –0.010 (0.127) |
| Depression | –0.008 (0.236) | –0.008 (0.240) |
| Self-consciousness | –0.012 (0.056) | –0.012 (0.056) |
| Impulsiveness | 0.002 (0.794) | 0.002 (0.802) |
| Vulnerability | –0.011 (0.059) | –0.011 (0.063) |
| Extraversion | 0.017 (0.022) | 0.017 (0.021) |
| Warmth | 0.021 (0.0004) | 0.021 (0.0004) |
| Gregariousness | 0.005 (0.483) | 0.005 (0.476) |
| Assertiveness | 0.002 (0.828) | 0.001 (0.862) |
| Activity | 0.001 (0.857) | 0.001 (0.864) |
| Excitement seeking | 0.010 (0.183) | 0.010 (0.165) |
| Positive emotions | 0.014 (0.020) | 0.015 (0.017) |
| Openness | 0.014 (0.035) | 0.014 (0.036) |
| Fantasy | 0.017 (0.016) | 0.017 (0.016) |
| Aesthetics | 0.011 (0.098) | 0.011 (0.102) |
| Feelings | 0.007 (0.319) | 0.007 (0.312) |
| Actions | 0.008 (0.195) | 0.008 (0.192) |
| Ideas | 0.009 (0.189) | 0.009 (0.186) |
| Values | 0.001 (0.887) | 0.001 (0.910) |
| Agreeableness | 0.017 (0.010) | 0.017 (0.011) |
| Trust | 0.006 (0.313) | 0.006 (0.321) |
| Straightforwardness | 0.012 (0.050) | 0.012 (0.055) |
| Altruism | 0.012 (0.042) | 0.012 (0.041) |
| Compliance | 0.008 (0.175) | 0.008 (0.179) |
| Modesty | 0.005 (0.471) | 0.005 (0.499) |
| Tendermindedness | 0.013 (0.025) | 0.013 (0.024) |
| Conscientiousness | 0.006 (0.409) | 0.005 (0.428) |
| Competence | 0.001 (0.846) | 0.001 (0.857) |
| Order | 0.003 (0.606) | 0.003 (0.637) |
| Dutifulness | 0.006 (0.325) | 0.006 (0.346) |
| Achievement striving | 0.005 (0.471) | 0.005 (0.442) |
| Self-discipline | 0.001 (0.854) | 0.001 (0.908) |
| Deliberation | 0.005 (0.414) | 0.005 (0.417) |

**Appendix 1—table 6.** Effect of mitochondrial haplogroups on mitochondrial DNA copy number (mtDNAcn).

| Haplogroups | Frequency (%) | Beta | SE | p-Value |
|---|---|---|---|---|
| N | 1.94 | −0.45 | 0.27 | 0.095 |
| H | 41.83 | −0.12 | 0.08 | 0.103 |
| T | 9.42 | 0.20 | 0.13 | 0.122 |
| X | 1.94 | −0.31 | 0.27 | 0.258 |
| K | 9.70 | 0.12 | 0.13 | 0.361 |
| W | 2.49 | 0.19 | 0.24 | 0.418 |
| V | 3.88 | −0.13 | 0.19 | 0.500 |
| U | 12.60 | 0.07 | 0.11 | 0.504 |
| I | 2.77 | −0.09 | 0.23 | 0.703 |
| J | 11.50 | 0.03 | 0.12 | 0.795 |

