## [Editor Report]

This paper makes a comprehensive survey of the relationship between mtDNAcn and the personality dimensions, as well as how and whether they mediate the relationships between personality dimensions and mortability as well as other behavioural measures that may lead to mortality.

---

## [Decision Letter]

**Decision letter after peer review:**

Thank you for submitting your article "Personality traits are consistently associated with blood mitochondrial DNA copy number estimated from genome sequences in two genetic cohort studies" for consideration by *eLife*. Your article has been reviewed by 2 peer reviewers, and the evaluation has been overseen by a Reviewing Editor and Ma-Li Wong as the Senior Editor. The following individual involved in review of your submission has agreed to reveal their identity: Na Cai (Reviewer #2).

Essential revisions:

1. mtDNAcn quantification: What are the WGS coverage for BLSA and SardiNIA respectively? It would be great to see a distribution of WGS coverage (over autosomes and over mtDNA, separately) in both studies, to get a sense of whether they are comparable, since quantification of mtDNAcn will be more accurate with higher WGS coverage. I suggest including these plots as additional supplementary figures and stating the coverage in main text.

2. Phenotype measures and definition: the authors wrote that they averaged across all four assessments of NEO-PI-R scores for BLSA participants as this "improves the BLSA analysis by reducing the standard deviation of the effects estimates and therefore increases statistical power". I suggest the authors show test-retest stability of these measures in a supplementary table or figure, drawing special attention to the age progression between the four assessments.

3. Personality mortality index (PMI) and mortality: In methods, the authors wrote that they computed a PMI based on "four NEO PI-R facets (vulnerability, activity, self-discipline, and competence) that have been previously shown as significantly associated with mortality", and "assigned a score of 0 or 1 to each individual based on the median value and the direction of the association of the trait with mortality". Is the association of the trait with mortality referred to here from reference 25? The authors also write in the main text that "we confirmed the association between PMI and mortality by finding that the participants had significantly longer survival time with increasing PMI from zero to four (Figure 4)". In Figure 4, I don't understand what the X axis means – it is labelled "Time" and goes from 0 to 6000, what units are these? Also, this means the authors must have derived a measure of mortality in BLSA and SardiNIA data, as they use it subsequently to perform mediation analysis between PMI, mtDNAcn and mortality. Is this mortality measure in survival time (ie. age at death, in the units shown in Figure 4)? Please explain what is "mortality" in all analysis that uses this measure, I can't find its derivation in the methods section. Also, please include a figure or table on the associations between each of the four NEO PI-R facets (vulnerability, activity, self-discipline, and competence) with "mortality" defined in BLSA and SardiNIA.

4. Mediation between PMI and mortability by mtDNAcn: Figure 4 legend writes "effect of PMI on mortality is fully mediated via mtDNAcn". I do not think this is true, as after regressing out mtDNAcn there is still significant direct effect of PMI to mortality. I suggest the authors change the phrasing of this to simply "mtDNAcn significantly mediates relationship between PMI and mortability".

5. Mediation of PMI on the relationship between mtDNAcn and mortality: I would have rather liked to see the mediation analysis performed the other way round. The current analysis tests for the indirect effect of PMI on mortality through mtDNAcn. I would like to see what are the direct and indirect effects of mtDNAcn on mortality, the latter controlling for PMI. This would allow us to better understand the relationship between PMI, mtDNAcn and mortality, and better infer the underlying biological mechanisms.

6. Previous studies have shown that mtDNAcn are potentially mediated by hormonal levels and thus menopause. Given the mean age of 57 in the SardiNIA cohort, the authors should investigate in more detail the potential confounding effects of menopause in women.

7. The only personality trait (out of the big five) available in the UK Biobank is neuroticism. Since the authors found that most of their associations are significant for this complex, I would strongly suggest they try to replicate their findings in patients from the UK Biobank which have both, genome-wide sequencing data as well the summary score of neuroticism (Data-Field 20127).

8. The amount of mtDNA varies across populations and across different haplogroups. The authors should therefore compute the major haplogroups present in Europeans and adjust/account for those variables in the correlation and mortality analyses.

9. The current study used buffy cells for sequencing, therefor the estimation of mtDNA per cell is not possible. Hence, the formula needs to be changed accordingly to indicate that the overall mtDNA abundance in blood is measured.

10. It is not clear which sequence coverage the authors want to include in their models. Autosomal, mitochondrial coverage or the total number of reads for each individual? Please specify.

11. It is not clear which regression method was used to assess mortality. I assume it was Cox regression adjusted for the most important covariates associated with mortality? This needs to be clearly stated.

12. Figure 4: Which regression coefficients are shown here? Those from a linear regression model? In this Figure, you also should show the 95% CI of the effect sizes and the P-Values as actual P-values not just asterisks (which are also not mentioned what they mean in the Figure legends).

*Reviewer #1 (Recommendations for the authors):*

Methods:

– mtDNA copy number estimation. The current study used buffy cells for sequencing, therefor the estimation of mtDNA per cell is not possible. Hence, the formula needs to be changed accordingly to indicate that the overall mtDNA abundance in blood is measured.

– Statistical analyses. It is not clear which sequence coverage the authors want to include in their models. Autosomal, mitochondrial coverage or the total number of reads for each individual? Please specify.

– Statistical analyses. It is not clear which regression method was used to assess mortality. I assume it was Cox regression adjusted for the most important covariates associated with mortality? This needs to be clearly stated.

Results:

– Figure 4: Which regression coefficients are shown here? Those from a linear regression model? In this Figure, you also should show the 95% CI of the effect sizes and the P-Values as actual P-values not just asterisks (which are also not mentioned what they mean in the Figure legends).

*Reviewer #2 (Recommendations for the authors):*

1. mtDNAcn quantification: What are the WGS coverage for BLSA and SardiNIA respectively? It would be great to see a distribution of WGS coverage (over autosomes and over mtDNA, separately) in both studies, to get a sense of whether they are comparable, since quantification of mtDNAcn will be more accurate with higher WGS coverage. I suggest including these plots as additional supplementary figures, and stating the coverage in main text.

2. Phenotype measures and definition: the authors wrote that they averaged across all four assessments of NEO-PI-R scores for BLSA participants as this "improves the BLSA analysis by reducing the standard deviation of the effects estimates and therefore increases statistical power". I suggest the authors show test-retest stability of these measures in a supplementary table or figure, drawing special attention to the age progression between the four assessments.

3. Personality mortality index (PMI) and mortality: In methods, the authors wrote that they computed a PMI based on "four NEO PI-R facets (vulnerability, activity, self-discipline, and competence) that have been previously shown as significantly associated with mortality", and "assigned a score of 0 or 1 to each individual based on the median value and the direction of the association of the trait with mortality". Is the association of the trait with mortality referred to here from reference 25? The authors also write in the main text that "we confirmed the association between PMI and mortality by finding that the participants had significantly longer survival time with increasing PMI from zero to four (Figure 4)". In Figure 4, I don't understand what the X axis means – it is labelled "Time" and goes from 0 to 6000, what units are these? Also, this means the authors must have derived a measure of mortality in BLSA and SardiNIA data, as they use it subsequently to perform mediation analysis between PMI, mtDNAcn and mortality. Is this mortality measure in survival time (ie. age at death, in the units shown in Figure 4)? Please explain what is "mortality" in all analysis that uses this measure, I can't find its derivation in the methods section. Also, please include a figure or table on the associations between each of the four NEO PI-R facets (vulnerability, activity, self-discipline, and competence) with "mortality" defined in BLSA and SardiNIA.

4. Mediation between PMI and mortability by mtDNAcn: Figure 4 legend writes "effect of PMI on mortality is fully mediated via mtDNAcn". I do not think this is true, as after regressing out mtDNAcn there is still significant direct effect of PMI to mortality. I suggest the authors change the phrasing of this to simply "mtDNAcn significantly mediates relationship between PMI and mortability".

5. Mediation of PMI on the relationship between mtDNAcn and mortality: I would have rather liked to see the mediation analysis performed the other way round. The current analysis tests for the indirect effect of PMI on mortality through mtDNAcn. I would like to see what are the direct and indirect effects of mtDNAcn on mortality, the latter controlling for PMI. This would allow us to better understand the relationship between PMI, mtDNAcn and mortality, and better infer the underlying biological mechanisms.

---

## [Author Response]

Essential revisions:1. mtDNAcn quantification: What are the WGS coverage for BLSA and SardiNIA respectively? It would be great to see a distribution of WGS coverage (over autosomes and over mtDNA, separately) in both studies, to get a sense of whether they are comparable, since quantification of mtDNAcn will be more accurate with higher WGS coverage. I suggest including these plots as additional supplementary figures and stating the coverage in main text.

We thank the reviewer for this comment. The average WGS coverage for BLSA and SardiNIA participants is 35.8X and 3.8X, respectively (because BLSA performed high coverage sequencing while SardiNIA did low-pass sequencing). We have included as Appendix 1-figure 7, four histograms describing the autosomal and mtDNA coverage for BLSA and SardiNIA. We have also modified the sentence in line 251 as shown below to include the sequence coverage in the main text.

“After adjusting for the effect of sex, sequence coverage (mean sequence coverage of 35.8X and 3.8X, respectively for BLSA and SardiNIA), platelet count, and white blood cell parameters, there remained a significant, albeit modest, inverse association of mtDNAcn with age in both study cohorts (Figure 1 and Appendix 1-table 1).”

2. Phenotype measures and definition: the authors wrote that they averaged across all four assessments of NEO-PI-R scores for BLSA participants as this "improves the BLSA analysis by reducing the standard deviation of the effects estimates and therefore increases statistical power". I suggest the authors show test-retest stability of these measures in a supplementary table or figure, drawing special attention to the age progression between the four assessments.

We thank the reviewer for the suggestion. For the BLSA cohort, there were four personality assessment points on average per participant. We used the average across all assessment points. We had stated the test-retest stability for the five personality domains for BLSA in the main text (line 162). Now, motivated by the reviewer’s suggestion, we have performed the test-retest stability test for all the facets and included the bar plot of the results as Appendix 1 -figure 5.

3. Personality mortality index (PMI) and mortality: In methods, the authors wrote that they computed a PMI based on "four NEO PI-R facets (vulnerability, activity, self-discipline, and competence) that have been previously shown as significantly associated with mortality", and "assigned a score of 0 or 1 to each individual based on the median value and the direction of the association of the trait with mortality". Is the association of the trait with mortality referred to here from reference 25?

Yes. We have made it clear in the main text as below (line 217):

“In subsequent analyses, we computed a personality-mortality index (PMI) using data from the four NEO PI-R facets (vulnerability, activity, self-discipline, and competence) that have been previously shown as significantly associated with mortality by Chapman et al. (25).”

The authors also write in the main text that "we confirmed the association between PMI and mortality by finding that the participants had significantly longer survival time with increasing PMI from zero to four (Figure 4)". In Figure 4, I don't understand what the X axis means – it is labelled "Time" and goes from 0 to 6000, what units are these?

The X-axis labeled Time had units in days, so the range of the time on the X-axis was from 0 to 6,000 days (0 – 16.4 years). We realized that using “day” as the unit is confusing and less interpretable. We have now updated the plot in figure 4 to use “year” as the time unit.

Also, this means the authors must have derived a measure of mortality in BLSA and SardiNIA data, as they use it subsequently to perform mediation analysis between PMI, mtDNAcn and mortality. Is this mortality measure in survival time (ie. age at death, in the units shown in Figure 4)?

We agree with the reviewer that we had not made this clear. Mortality used in the mediation analysis is a binary variable describing the dead or alive status of study participants. We have made this clearer in the main text in line 230:

“Finally, we used R package lavaan (version 0.6-7) to perform a mediation analysis to test whether mtDNAcn mediates the association between personality (i.e., PMI) and mortality risk (a binary variable describing the dead or alive status of study participants).”

Please explain what is "mortality" in all analysis that uses this measure, I can't find its derivation in the methods section. Also, please include a figure or table on the associations between each of the four NEO PI-R facets (vulnerability, activity, self-discipline, and competence) with "mortality" defined in BLSA and SardiNIA.

We thank the reviewer for this comment. We used mortality data to derive time-to-event in the Kaplan-Meier plot shown in figure 4. Mortality in subsequent analyses is a binary variable describing the dead or alive status of study participants. We had stated this in the main text line 228.

Also, as suggested by the reviewer, we have performed a Cox Proportional Hazards analysis for the four NEO PI-R facets and included the results as Appendix 1-table 3. We have also summarized the results in the main text in line 325:

“First, we performed a Cox Proportional Hazards analysis for the four NEO PI-R facets (Appendix 1-table 3). The analysis showed that after adjusting for age and sex, all four facets are significantly associated with increased risk of mortality (vulnerability) or reduced risk of mortality (activity, self-discipline, and competence).”

4. Mediation between PMI and mortability by mtDNAcn: Figure 4 legend writes "effect of PMI on mortality is fully mediated via mtDNAcn". I do not think this is true, as after regressing out mtDNAcn there is still significant direct effect of PMI to mortality. I suggest the authors change the phrasing of this to simply "mtDNAcn significantly mediates relationship between PMI and mortability".

We appreciate this comment, and have changed the wording of that figure legend to reflect the reviewer’s comment:

“The effect of Personality-mortality Index (PMI) on mortality is significantly mediated via mtDNAcn.”

5. Mediation of PMI on the relationship between mtDNAcn and mortality: I would have rather liked to see the mediation analysis performed the other way round. The current analysis tests for the indirect effect of PMI on mortality through mtDNAcn. I would like to see what are the direct and indirect effects of mtDNAcn on mortality, the latter controlling for PMI. This would allow us to better understand the relationship between PMI, mtDNAcn and mortality, and better infer the underlying biological mechanisms.

We thank the reviewer for this suggestion. We have performed a mediation analysis that treats PMI as the mediator and tests the direct and indirect effect of mtDNAcn on mortality. We have included the results as Appendix 1-table 4. We have also summarized the results in the supplementary information:

“We used causal mediation analysis to test the indirect effect of mtDNAcn on mortality via PMI as a mediator. The indirect effect estimate showed that the effect of mtDNAcn on mortality is not mediated through PMI. We tested for the significance of the indirect effect using 5000 bootstrap samples and it was not statistically significant (p-values = 0.355 and 0.963, respectively for BLSA and SardiNIA). See Appendix 1-table 4 for more details.”

6. Previous studies have shown that mtDNAcn are potentially mediated by hormonal levels and thus menopause. Given the mean age of 57 in the SardiNIA cohort, the authors should investigate in more detail the potential confounding effects of menopause in women.

We thank the reviewer for this suggestion. In response, we have investigated menopausal status as a potential confounder on mtDNAcn. We found that there was no significant difference between the mtDNAcn of pre-menopausal and menopausal women (p-value = 0.56; a plot as shown in Appendix 1-figure 9). The results show that menopause had no effect on mtDNAcn in the SardiNIA cohort. Also, we ran two association models between mtDNAcn and NEO PI-R traits in women, with and without adjustment for the effect of menopause status (both models adjusted for the effect of age, sequence coverage, WBC, platelet count, and percentages of major types of leukocytes). The results are shown in the supplementary information in Appendix 1-table 5. The results show that menopause status does not affect the association between mtDNAcn and personality: the effect sizes and p-values were almost the same in both models for all the traits.

7. The only personality trait (out of the big five) available in the UK Biobank is neuroticism. Since the authors found that most of their associations are significant for this complex, I would strongly suggest they try to replicate their findings in patients from the UK Biobank which have both, genome-wide sequencing data as well the summary score of neuroticism (Data-Field 20127).

We thank the reviewer and agree that a further extension of our findings to a larger cohort like the UK Biobank would be a natural next step for this study. However, the current study already provides replication of the results in two independent cohorts, and thus we prefer to retain the focus on these cohorts. The reasons include:

1) We have found the association between mtDNAcn and neuroticism traits in two independent cohorts for which neuroticism traits were measured in exactly the same way (using the NEO PI-R Inventory with 48 items). In contrast, the UK Biobank participants were assessed for neuroticism with 12 items, and the measurements were therefore different. Furthermore, the UKB did not assess the other personality factors or facets, and we could not perform most of the analyses presented in this manuscript.

2) An analysis of UK Biobank data in this current study is a logistical challenge. In addition to gaining access to UK Biobank data, we would also have to calculate the mtDNAcn for UK Biobank participants with whole-genome sequencing data (currently for 200,000 participants) – which would be computationally expensive and require considerable time and effort by staff who are now differently tasked.

Therefore, we believe that the replication in two independent cohorts meets the criterion for publication of a first report.

Still, we strongly agree with the Reviewer that replication in the UK Biobank is important, and we have added the following paragraph in the Discussion section line 465:

“In our current study, we showed a significant association between mtDNAcn and neuroticism traits that replicated in two independent cohorts. Other larger cohorts (e.g., UK Biobank) have also collected certain personality information (e.g., questions about neuroticism) as well as whole-genome sequencing data (which makes estimating mtDNAcn feasible). Therefore, trying to replicate our findings in such larger cohorts will be a natural extension of this study.”

8. The amount of mtDNA varies across populations and across different haplogroups. The authors should therefore compute the major haplogroups present in Europeans and adjust/account for those variables in the correlation and mortality analyses.

We thank the reviewer for this comment. We have computed the major haplogroups present in our dataset. There were 17 haplogroups in total, and we excluded seven haplogroups that were extremely rare (present in less than 5 study participants). The most common haplogroup was H (Frequency of about 42%). The boxplot shows the relative mtDNAcn in each haplogroup. mtDNAcn has a similar distribution among the different haplogroups, although slight variations in mtDNAcn were observed amongst them. We further investigated if these mtDNAcn variations amongst the haplogroups were statistically significant by testing for the association of each haplogroup versus all other haplogroups with mtDNAcn. Amongst the 10 haplogroups present in more than 5 study participants, none was significantly associated with mtDNAcn. This finding suggests that none of the haplogroups present in our study participants will impact our results. Our results are also consistent with the literature. For example, Cai and colleagues stated in their paper (Cai et al., *Front Genet*. 2020) that “Testing each haplogroup against all others for association with mtDNA copy number (Table 1), we found that mtDNA copy number is significantly higher in Haplogroup L, and lower in Haplogroups A, B, C, D, and F, none of which occur at high frequencies in Europe.” Indeed, the 6 haplogroups that were significantly associated with mtDNA copy number were also very rare in our study: L present in 4 participants; A present in 2 participants; B present in 1 participant; C present in 2 participants; D present in 3 participants; and F present in 0 participants. We have included the figure and table as supplementary information (Appendix 1-figure 10 and Appendix 1-table 6).

9. The current study used buffy cells for sequencing, therefor the estimation of mtDNA per cell is not possible. Hence, the formula needs to be changed accordingly to indicate that the overall mtDNA abundance in blood is measured.

We thank the reviewer for this important comment. We indeed used DNA extracted from buffy coat, which is the most common DNA source for sequencing, to do the whole-genome sequencing analyzed here. Buffy coat contains leukocytes and platelets, so the reviewer is certainly right that the mtDNA copy number is not estimated for a specific cell type.

Without considering platelets, the formula for mtDNA copy number calculates a weighted average (weights determined by the proportion of the leukocytes populations) of mtDNA copy number per cell across all leukocytes; each cell has two copies of nuclear DNA that provide the reference for the calculation. The platelet content of the buffy coat then makes the calculation somewhat more complicated and less precise. Because platelets have mtDNA but do not contain nuclear DNA, the formula overestimates the mtDNA copy number (Rausser et al. *eLife* 2021). One platelet contains on average 1.6 molecules of mtDNA (Urata et al. Ann Clin Biochem 2008; Hurtado-Roca et al. PLoS One 2016). And there are on average 40 platelets per leukocyte in whole blood (Urata et al. Ann Clin Biochem 2008), though that number is likely to be substantially lower in the buffy coat (we have no precise measurements of the amounts). Thus, the impact of platelets on buffy coat mtDNA copy number is likely small but certainly not negligible. This represents a limitation of both our own and every other study of mtDNA in blood DNA material (Picard M. Mitochondrion 2022).

Because of this limitation, papers published in the field typically use both platelet count, and white blood cell counts as covariates in any models testing the association between mtDNA copy number and quantitative traits. We also used this conventional approach in all of our analyses.

In response to the important comment of the Reviewer, we have added the following paragraph to the Discussion section line 450:

“Additionally, we note that the formula for mtDNA copy number calculates a weighted average (weights determined by the proportion of leukocyte populations) of mtDNA copy number per cell across all leukocytes; each cell has two copies of nuclear DNA that provide the reference for the calculation. However, this does not take into account the platelet content of the buffy coat. Because each platelet has about 1.6 copies of mtDNA (Urata et al. Ann Clin Biochem 2008; Hurtado-Roca et al. PLoS ONE 2016) but does not contain nuclear DNA, the formula somewhat overestimates mtDNA copy number (Rausser et al. *eLife* 2021). The platelet content of buffy coat may also vary, representing a limitation of this and other studies of mtDNA in blood DNA material (Picard M. Mitochondrion 2022). Because of this limitation, papers published on mtDNA copy number typically use both platelets count, and white blood cell counts as covariates in any models testing the association between mtDNA copy number and quantitative traits. We also used this conventional approach in our analyses.”

10. It is not clear which sequence coverage the authors want to include in their models. Autosomal, mitochondrial coverage or the total number of reads for each individual? Please specify.

We have now made it clear in the methods that we use autosomal coverage for each individual in the regression models by adding this sentence in line 187:

“…sequence coverage (the average autosomal coverage for each study participant),”

11. It is not clear which regression method was used to assess mortality. I assume it was Cox regression adjusted for the most important covariates associated with mortality? This needs to be clearly stated.

Yes, we used the Cox Proportional Hazards model and adjusted for age and sex. We have made it clearer in the main text by adding this sentence in line 213:

**“**We also used the Cox Proportional Hazards model to test for their association with mortality, adjusting for age and sex.”

12. Figure 4: Which regression coefficients are shown here? Those from a linear regression model? In this Figure, you also should show the 95% CI of the effect sizes and the P-Values as actual P-values not just asterisks (which are also not mentioned what they mean in the Figure legends).

The regression coefficients shown in figure 4 are the coefficients from the causal mediation analysis. We have updated the figure to show the 95% CI and p-values of the coefficients and have also been explicit in the figure legend.